# Dissecting the genetic landscape of GPCR signaling through phenotypic profiling in *C. elegans*

Longjun Pu[1,2,3,12], Jing Wang[1,2,3,12], Qiongxuan Lu[1,2,3], Lars Nilsson[1,2,3], Alison Philbrook[4], Anjali Pandey[4], Lina Zhao[1,2,3], Robin van Schendel [5], Alan Koh [6,7], Tanara V. Peres [6,7], Weheliye H. Hashi[6,7], Si Lhyam Myint[1,8,9], Chloe Williams[10], Jonathan D. Gilthorpe [10], Sun Nyunt Wai [1,8,9], Andre Brown[6,7], Marcel Tijsterman [5], Piali Sengupta [4], Johan Henriksson [1,8,11] ✉ & Changchun Chen [1,2,3] ✉

G protein-coupled receptors (GPCRs) mediate responses to various extracellular and intracellular cues. However, the large number of GPCR genes and their substantial functional redundancy make it challenging to systematically dissect GPCR functions in vivo. Here, we employ a CRISPR/Cas9-based approach, disrupting 1654 GPCR-encoding genes in 284 strains and mutating 152 neuropeptide-encoding genes in 38 strains in *C. elegans*. These two mutant libraries enable effective deorphanization of chemoreceptors, and characterization of receptors for neuropeptides in various cellular processes. Mutating a set of closely related GPCRs in a single strain permits the assignment of functions to GPCRs with functional redundancy. Our analyses identify a neuropeptide that interacts with three receptors in hypoxia-evoked locomotory responses, unveil a collection of regulators in pathogen-induced immune responses, and define receptors for the volatile food-related odorants. These results establish our GPCR and neuropeptide mutant libraries as valuable resources for the *C. elegans* community to expedite studies of GPCR signaling in multiple contexts.

G protein-coupled receptors (GPCRs) represent the largest family of cell surface proteins in metazoa. GPCRs are expressed in diverse cell types, regulating a plethora of physiological and pathological processes[1,2]. The human genome encodes approximately 800 GPCRs, which cluster in five major classes based on sequence similarity and evolutionary relationships[3,4]. GPCRs are at the interface of environmental stimuli and intracellular responses, translating external signals into internal representations. They share a basic common structure of seven transmembrane domains, but display remarkable diversity in their ligand binding properties, intracellular signal transduction, and physiological functions[4]. The expansion of the GPCR family, coupled with a high degree of functional redundancy, creates

[1]Department of Molecular Biology, Umeå University, Umeå, Sweden. [2]Umeå Centre for Molecular Medicine, Umeå University, Umeå, Sweden. [3]Wallenberg Centre for Molecular Medicine, Umeå University, Umeå, Sweden. [4]Department of Biology, MS 008, Brandeis University, 415 South Street, Waltham, MA 02454, USA. [5]Department of Human Genetics, Leiden University Medical Center, Leiden, The Netherlands. [6]MRC Laboratory of Medical Sciences, London W12 0HS, UK. [7]Institute of Clinical Sciences, Imperial College London, London, UK. [8]Umeå Centre for Microbial Research (UCMR), Umeå University, Umeå, Sweden. [9]The Laboratory for Molecular Infection Medicine Sweden (MIMS), Umeå University, Umeå, Sweden. [10]Department of Integrative Medical Biology, Umeå University, Umeå, Sweden. [11]Integrated Science Lab (Icelab), Umeå University, Umeå, Sweden. [12]These authors contributed equally: Longjun Pu, Jing Wang. ✉e-mail: johan.henriksson@umu.se; changchun.chen@umu.se

substantial complexities in systematically dissecting their functions. In particular, for many GPCRs, it has been challenging to identify their physiologically relevant ligands. Rapid advances in high-throughput screening techniques, computational modeling, and structural analyses have contributed significantly to the understanding of GPCR activation and signaling[1,2,5]. However, these analyses have typically been performed in heterologous systems. A systematic in vivo approach to deorphanize the receptors and decipher their functions remains to be described.

The genome of the free-living nematode *C. elegans* encodes one of the largest GPCR repertoires among any sequenced organisms[6–9]. In *C. elegans*, GPCRs mediate or regulate chemosensation, nociceptive responses, lipid homeostasis, social behavior, immunity, and mating[10–12], allowing animals to thrive in a complex ecological niche. A subset of GPCRs in *C. elegans*, mainly neurotransmitter receptors, is highly conserved. By contrast, predicted chemoreceptor GPCRs are highly divergent[6,9,10] and dramatically expanded, numbering over 1300[9]. The chemoreceptors are expressed in a small nervous system consisting of 302 neurons, of which 32 are chemosensory[12]. Since each chemosensory neuron expresses multiple chemoreceptors[9,13,14], individual neurons likely detect and discriminate a variety of sensory stimuli to elicit the appropriate behavioral responses[11,12].

Many neuropeptide receptors in *C. elegans* have been deorphanized in vitro[15], facilitating the subsequent functional analyses.

However, the majority of chemoreceptors, as well as a proportion of neuropeptide receptors, remain fully uncharacterized. In particular, only a few chemoreceptors have been paired with defined ligands since the description of the first olfactory receptor ODR-10[12,16]. The large number of GPCRs in *C. elegans*, their functional redundancy, and the complexities associated with expressing these GPCRs in heterologous systems render existing approaches, such as forward and reverse genetics, inefficient in determining the functions of GPCRs in physiologically relevant contexts.

To this end, we sought to generate a comprehensive, versatile, and widely applicable resource that could overcome the current obstacles associated with the dissection of GPCR function in *C. elegans*. We used CRISPR/Cas9 genome editing to disrupt 1654 GPCR-encoding genes and 152 neuropeptide-encoding genes in 284 and 38 strains, respectively. To bypass possible functional redundancy, we specifically targeted multiple genes encoding closely related GPCRs in individual strains. We then systematically screened mutant strains for their responses to a range of environmental signals. Our analyses established a role for peptidergic signaling in acute response to hypoxia, identified a panel of GPCRs and neuropeptides involved in response to pathogens, and determined the putative receptors for the attractive volatile odorants pyrazine and 2,3-pentanedione (Fig. 1a). In particular, we identified GPCRs that exhibit partial redundancy in their functions, highlighting a distinctive advantage of our approach over existing

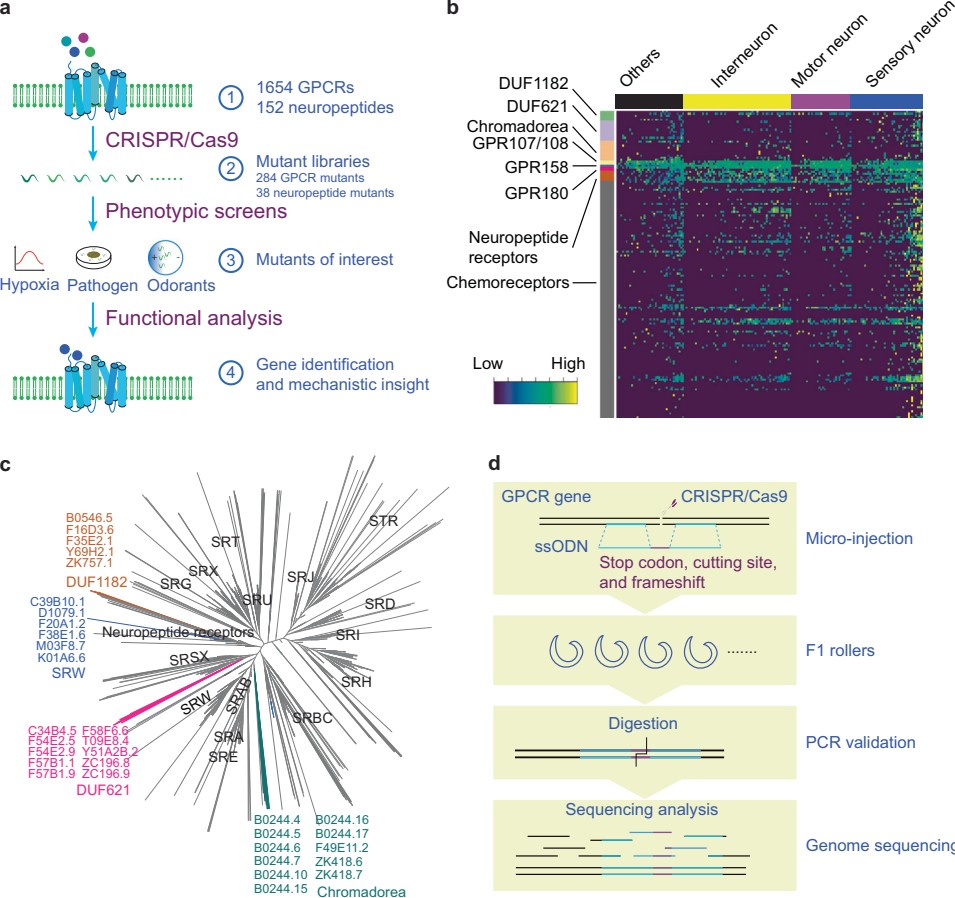

**Fig. 1 | The construction of GPCR and neuropeptide mutant libraries.**
**a** Schematic drawing of our approach to phenotypic profiling of nearly all GPCR genes in *C. elegans*. **b** The gene expression pattern of unannotated GPCR genes, according to a previously published single-cell RNA-seq dataset of L4 worms[13]. 'Others' indicates the non-neuronal tissues, such as intestine and muscle. **c** Phylogenetic tree analysis of 1675 GPCRs in *C. elegans*. The GPCR sub-families and relevant genes are highlighted. Six receptors of SRW subfamily, as indicated in blue, were clustered to the clade of neuropeptide receptors. 10 DUF621 domain-containing receptors (magenta) were closely related to chemoreceptors, and 5 DUF1182 domain-containing receptors (brown) were in the clade of neuropeptide receptors. 11 putative GPCRs (Teal), which were annotated as Chromadorea class in the Wormbase, were clustered closely to chemoreceptors. **d** Strategy for the generation and validation of GPCR mutant strains.

methods. We expect that our resource will streamline the analyses of GPCR function in *C. elegans*, leading to new insights into how GPCRs translate external information into intracellular responses.

## Results

### Comprehensive identification of GPCR encoding genes

The number of GPCRs encoded by the *C. elegans* genome is predicted to be more than 1300[6–9]. We reasoned that updated gene annotation might allow us to identify additional GPCR-encoding genes. From *C. elegans* genome release WS273 (and later versions), we obtained a list of 1442 putative chemoreceptor encoding genes, including 126 that had not been annotated (Supplementary Data 1a, b; See methods for details)[6–9]. Unannotated genes belong to multiple chemoreceptor subfamilies, and are predicted to be expressed predominantly in chemosensory neurons based on single-cell transcriptomics analysis[13] (Fig. 1b and Supplementary Data 1b, c). Six unannotated members of the *srw* family are closely related to neuropeptide receptors (Fig. 1c; Supplementary Data 1b)[6,7]. We further identified 10 DUF621 domain and 5 DUF1182 domain-containing proteins by searching for putative GPCRs within the families defined by the hidden Markov model GPCRHMM[17] (Supplementary Data 1a). Expression analyses showed an enrichment of DUF621 family genes in chemosensory neurons (Fig. 1b), and phylogenetic analysis suggested their close relationship to chemoreceptors (Fig. 1c). By contrast, DUF1182 domain-containing proteins were broadly expressed (Fig. 1b) and clustered with the clade of neuropeptide receptors (Fig. 1c). Additionally, we identified 11 nematode-specific GPCRs, which shared significant sequence similarity and were clustered closely with chemoreceptors (Fig. 1c and Supplementary Data 1a; annotated as GPCRs Chromadorea in Wormbase). Furthermore, we specifically sought for putative orthologs of human GPCRs, which led to the identification of C11H1.2 and Y75B8A.16 (GPR89), C15H9.5 and C52B9.4 (GPR107/GPR108), F39B2.8 (GPR158), and C15A7.2 and T04F8.2 (GPR180) (Supplementary Data 1a). These molecules have not been studied extensively in *C. elegans*, and have not been clustered into any major subfamilies. The GPR89 orthologs, Y75B8A.16 and C11H1.2, are more likely to be Golgi pH regulators[18], and are therefore named GPHR-1 and GPHR-2, respectively. We include them as non-GPCRs (Supplementary Data 1a). Together, we involved a set of 1675 putative GPCRs in this study (Supplementary Data 1a).

### Construction of GPCR and neuropeptide mutant libraries

We sought a genome editing strategy to systematically disrupt the GPCR genes, and opted to employ homology-directed integration of single-stranded oligodeoxynucleotide (ssODN), catalyzed by optimized ribonucleoprotein complexes (Fig. 1d)[19,20]. The insertion of ssODN simultaneously introduces stop codons while removing a short coding sequence to generate frameshifts (Fig. 1d and Supplementary Data 1d). The inclusion of a unique restriction enzyme cutting site facilitates tracking of editing events (Fig. 1d and Supplementary Data 1d). To reduce the number of microinjections involved, we typically disrupted three genes at a time. Multiple chemoreceptor genes may have arisen via gene duplication in *C. elegans* and thus share extensive sequence similarity[7,10]. To address redundant functions, we clustered the genes sharing sequence identity of >40% into one group[21]. Remaining genes were further assigned into the appropriate groups if they were closely linked to any genes within the existing groups on the same chromosome or in phylogenetic analyses. Other GPCR genes were placed arbitrarily into groups based on the numerical orders of gene names (See methods for details). In total, 1657 GPCR-encoding genes were assigned to 284 groups, and the genes within each group were disrupted in a single strain (Supplementary Data 1e). Our efforts led to the disruption of 1654 GPCR encoding genes, distributed across 284 strains (Supplementary Data 1d, e). The attempts to obtain homozygous mutants of *lin-17*, *mom-5*, or *lat-1* were unsuccessful,

and we have not yet pursued 18 genes on our list (Supplementary Data 1d).

To confirm gene disruption, we performed whole genome sequencing of 278 strains, bearing mutations in the total of 1628 GPCR-encoding genes and 29 non-GPCR encoding genes (Supplementary Data 1d, e). A set of genes were disrupted in multiple strains, yielding a total of 1725 alleles (Supplementary Data 1f). The editing information at the targeted sites is listed in Supplementary Data 1f, which does not include 26 genes disrupted after genome sequencing (Supplementary Data 1d). To complement the GPCR mutant library, we also disrupted 152 genes encoding neuropeptides in 38 strains (Supplementary Data 2). The creation of two mutant libraries allows us to simultaneously examine GPCR-mediated sensory perception and internal signal transduction. As proof of principle, we performed three screens to identify GPCRs and neuropeptides involved in response to acute hypoxia, pathogens, and volatile odorants (Fig. 1a).

### Peptidergic signaling is required for hypoxia-evoked locomotory response

*C. elegans* dramatically increases its locomotory speed when $O_2$ levels decrease rapidly from preferred (7%) to aversive hypoxia (1%)[22]. Double mutant animals for the G proteins *gpa-3* and *odr-3* are defective in locomotory response to acute drop of $O_2$ tension[22], implicating GPCR-regulation of this response. To test this notion, we assessed all GPCR mutants for their responses to acute hypoxia (Supplementary Data 3). One strain, in which five genes *dmsr-4*, *dmsr-5*, *dmsr-6*, *dmsr-7*, and *dmsr-8* were disrupted, showed defects in hypoxia-evoked arousal (Supplementary Fig. 1a). DMSRs are a class of peptide receptors related to the *Drosophila* myosuppressin receptors[6]. The quintuple mutant had a higher baseline speed at 7% $O_2$, and an attenuated locomotory response to 1% $O_2$ (Supplementary Fig. 1a). The response of *dmsr-4; dmsr-7; dmsr-8* triple mutants was comparable to that of the original quintuple strain (Fig. 2a), whereas no severe defects were observed as long as one of *dmsr-4*, *dmsr-7* or *dmsr-8* genes remained undisrupted (Supplementary Fig. 1b–h). The expression of either *dmsr-7* or *dmsr-8* under their endogenous promoters partially rescued the defect of the triple mutants (Fig. 2b and Supplementary Fig. 1i). These observations suggest that DMSR-4, 7 and 8 act redundantly to regulate acute response to hypoxia as well as basal locomotion under 7% $O_2$.

We next sought to determine which neuropeptide(s) signals through DMSR-4, 7 and 8 to modulate hypoxia-evoked locomotory response. Assay of all neuropeptide mutants identified one strain, displaying a response that was similar, but not identical, to that of *dmsr* mutants (Supplementary Fig. 1a, j). This strain, in which four genes *flp-1*, *flp-14*, *flp-23*, and *flp-25* were disrupted, had a strong post-hypoxia acceleration that was not evident in the receptor mutants (Supplementary Fig. 1a, j). Examining the responses of each single mutant revealed that mutating *flp-1* yielded a phenotype similar to that of the quadruple mutants (Fig. 2c and Supplementary Fig. 1k–m). The elevated basal locomotion of *flp-1* mutants has previously been reported[23,24], which was likely caused by altered neurotransmission at neuromuscular junctions[25]. The defects of *flp-1; dmsr* quadruple mutants were comparable to the receptor mutants (Fig. 2d), suggesting that FLP-1 might act through DMSR-4, 7 and 8 to maintain basal locomotion and regulate hypoxia-evoked behavioral response. This is supported, in part, by the documented interaction between FLP-1 and DMSR-7[15,26]. NPR-4, 5, 6, 11 and FRPR-7 have also been suggested as the receptors for FLP-1[15,27,28], but did not appear to participate in acute response to hypoxia (Supplementary Data 3). We next aimed to determine the specific neurons in which FLP-1 acts. Expressing *flp-1* cDNA specifically in AVK, but not in the other *flp-1* expressing neurons[24], partially rescued the defect of *flp-1* mutants (Fig. 2e and Supplementary Fig. 1n, o), suggesting that the release of FLP-1 from AVK neurons is crucial for behavioral response to acute hypoxia. Collectively, our GPCR and neuropeptide mutant libraries allowed us to

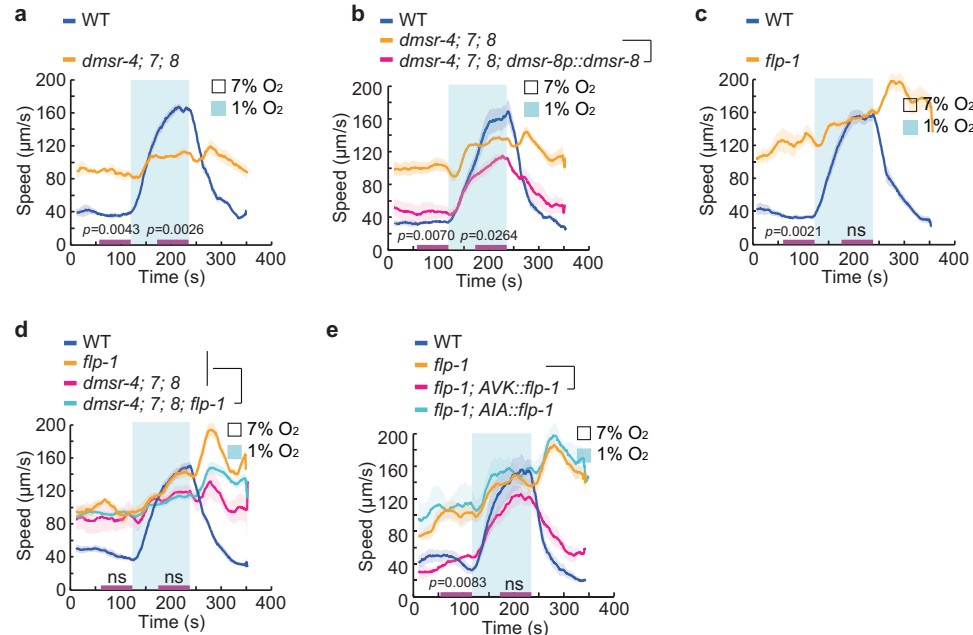

**Fig. 2 | Peptidergic signaling is required for hypoxia-evoked locomotory responses. a** Locomotory responses to rapid shifts from 7% to 1% $O_2$ of animals with indicated genotypes WT (N2) and *dmsr-4(yum5084); dmsr-7(yum5085); dmsr-8(yum5086)* triple mutants. In this and subsequent figure panels, purple bars on X-axis indicate the time intervals used for statistical analysis. $n = 3$ independent assays for each genotype. Data are presented as mean values +/- SEM. *p* values are displayed in the plot. Two-sided Welch's *t* test. **b** Locomotory responses to rapid shifts from 7% to 1% $O_2$ of animals with indicated genotypes WT, *dmsr-4(yum5084); dmsr-7(yum5085); dmsr-8(yum5086)* triple mutants, and transgenic triple mutants expressing *dmsr-8* cDNA from its endogenous promoter. $n = 3$ independent assays for each genotype. Data are presented as mean values +/- SEM. *p* values are displayed in the plot. Two-sided Welch's *t* test. **c** Locomotory responses to rapid shifts from 7% to 1% $O_2$ of animals with indicated genotypes WT and *flp-1(yum104)* mutants. $n = 4$ independent assays for each genotype. Data are presented as mean values +/- SEM. *p* values are displayed in the plot. ns = not significant (1% $O_2$). Two-sided Welch's *t* test. **d** Locomotory responses to rapid shifts from 7% to 1% $O_2$ of animals with indicated genotypes WT, *flp-1(yum104)*, *dmsr-4(yum5084); dmsr-7(yum5085); dmsr-8(yum5086)* triple, and *dmsr-4(yum5084); dmsr-7(yum5085); dmsr-8(yum5086); flp-1(yum5088)* quadruple mutants. $n = 4$ independent assays for each genotype. Data are presented as mean values +/- SEM. ns = not significant. Two-sided Welch's *t* test. **e** Locomotory responses to rapid shifts from 7% to 1% $O_2$ of animals with indicated genotypes WT, *flp-1(yum104)*, and transgenic *flp-1(yum104)* expressing *flp-1* cDNA from a truncated version of *flp-1* promoter, which drives the gene expression specifically in AVK neurons, and from *gcy-28.d* promoter in AIA neurons. $n = 3$ independent assays for each genotype except $n = 4$ for *flp-1(yum104)*. Data are presented as mean values +/- SEM. *p* values are displayed in the plot. ns = not significant (1% $O_2$). Two-sided Welch's *t* test.

## Identification of regulators in defense response to bacterial pathogen infection

Several GPCRs have been implicated in regulating *C. elegans'* defensive responses to pathogens[29–41]. To identify additional regulators, we screened the GPCR and neuropeptide mutants for their responses to the gram-negative pathogen *Vibrio cholerae*. We first explored if *V. cholerae* infection triggered cellular responses in *C. elegans* using RNA sequencing (RNA-seq). Exposing wild-type animals to *V. cholerae* for 8 hours significantly reprogramed gene expression, such that 407 genes were downregulated and 551 genes were upregulated (Supplementary Fig. 2a, b and Supplementary Data 4a, for adj. $p < 1e-20$). These genes showed a substantial overlap with the list of genes previously reported as being induced by *Pseudomonas aeruginosa* (Fig. 3a; Supplementary Fig. 2a, b)[42]. Gene Ontology (GO) analysis revealed that defense response to bacterial pathogen and fatty acid metabolism were the most upregulated and repressed cellular processes by *V. cholerae* (Supplementary Fig. 2c, d), similar to the patterns induced by *P. aeruginosa*[42,43]. However, the mechanisms underlying the slow killing of *C. elegans* by *V. cholerae* and *P. aeruginosa* are distinct. The virulence of *V. cholerae* to *C. elegans* involves a secreted cytotoxin MakA[44]. Expressing MakA encoding operon in the non-pathogenic *E.* *coli* strain Top10 was sufficient to render this bacterium virulent to *C. elegans* (Supplementary Fig. 2e). Therefore, using *V. cholerae* as an infection model could not only explore general principles of the host response to bacterial infection, but also identify unique regulators of the immune response to this pathogen.

We exposed GPCR and neuropeptide mutants to a partial lawn of *V. cholerae*, and monitored behavioural avoidance and survival in the same worm population. A set of mutants exhibited defects in escape from *V. cholerae*, and we prioritized 11 strains with the strongest phenotypes for further analysis (Fig. 3b, c; Supplementary Fig. 2f; Supplementary Data 4b). In 10 strains defective in pathogen avoidance, we confirmed the functional importance of 9 genes, including *npr-1*, *flp-21*, *fmi-1*, *pdfr-1*, *pdf-1*; *pdf-2*, *dmsr-7*, *dop-6*, and *F59B2.13* (Fig. 3c, d; Supplementary Fig. 2g–n). These mutants had normal responses to hypoxia-evoked food leaving (Supplementary Fig. 2o)[45], partially excluding that locomotory defects underlie the attenuated pathogen avoidance. Among the identified regulators, we noted the presence of ligand/receptor pairs, FLP-21/NPR-1, and PDF-1, PDF-2/ PDFR-1, as well as FLP-1/DMSR-7 (Supplementary Fig. 2p). These data further highlight the potential of our resource in determining ligand/receptor interactions in the relevant cellular processes. The contribution of PDFR-1 signaling appeared to be specific for *V. cholerae* since it was not involved in avoiding *P. aeruginosa* strain PA14[46]. DOP-6, F59B2.13 and FMI-1, which have not been previously implicated in behavioral response to bacterial pathogens, are broadly expressed in the nervous system[13] and may serve as the regulators of neuronal responses to bacterial infection.

effectively identify the ligand/receptor pairs involved in a complex behavior. These data also highlight the potential of our resource to identify functionally redundant genes, which would be difficult to characterize through alternative approaches.

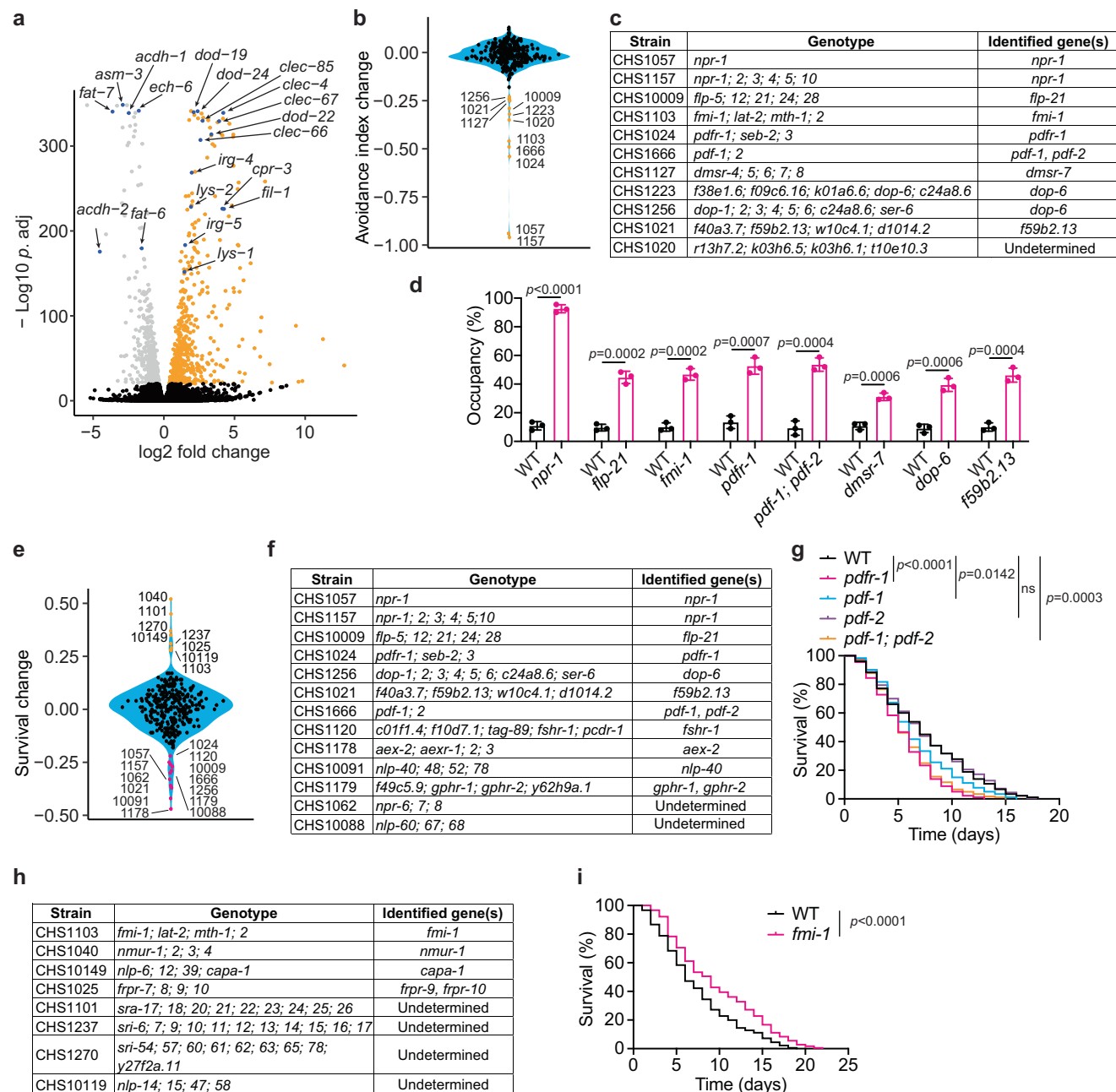

**Fig. 3 | Seeking GPCR and neuropeptide mutants that are defective in response to *V. cholerae*. a** Volcano plot showing the differentially expressed genes with adjusted *p* < 1e-20 in L4 animals exposed to *V. cholerae* for 8 hours. A set of genes involved in defense against bacterial infection and fatty acid metabolism were highlighted in blue. Random variations (jitter) were added to adjusted *p* < 1e-300. Two-sided Wald test. **b** Violin plot of avoidance index changes of the mutants relative to WT after 24-hour on *V. cholerae*. Mutants with severe defects were highlighted in orange. **c** The genotype of each strain indicated in (**b**) and the confirmed genes in those strains were listed in the table. **d** The proportion of animals on pathogen lawn after 24 hours of *V. cholerae* exposure. *n* = 3 biological replicates. Data are presented as mean values +/- SEM. *p* values are displayed in the plot. Two-tailed *t* test. **e** Violin plot of mean survival changes of mutants relative to WT.

Hypersensitivity mutants were marked in magenta, and resistant mutants were indicated in orange. **f** The genotypes of hypersensitive mutants to *V. cholerae* as indicated in magenta in (**e**). The genes required for survival on *V. cholerae* were identified in all strains except CHS1062 and CHS10088. **g** Survival of WT, *pdfr-1(yum2896)*, *pdf-1(yum2897)*, *pdf-2(yum2898)* and *pdf-1(yum2897); pdf-2(yum2898)* on *V. cholerae*. *n* = 2 biological replicates. *p* values are displayed in the plot. log-rank test. **h** The genotypes of resistant mutants to *V. cholerae* as indicated in orange in (**e**). The relevant genes in the strains CHS1025, CHS1040, CHS1103, and CHS10149 have been identified. **i** Survival of WT and *fmi-1(yum2825)* upon exposure to *V. cholerae*. *n* = 2 biological replicates. *p* values are displayed in the plot. log-rank test. Source data are provided in the Source Data file.

When animals were assayed on a partial lawn of pathogen, increased dwelling on the lawn generally correlated with enhanced susceptibility (Supplementary Fig. 3a). As expected, nearly all mutants defective in pathogen avoidance were susceptible to *V. cholerae* (Fig. 3e, f; Supplementary Fig. 3a, b; Supplementary Data 4c). Of the 13 most pathogen-sensitive strains, more than half exhibited avoidance

defects, including *npr-1*, *flp-21*, *pdfr-1*, *pdf-1; pdf-2*, *dop-6*, and *F59B2.13* mutants (Fig. 3d–g; Supplementary Fig. 3b–h). Subsequent analyses identified 5 additional genes, *fshr-1*, *aex-2*, *nlp-40*, *gphr-1*, and *gphr-2*, that were required for animals' survival following pathogen exposure (Fig. 3f; Supplementary Fig. 3i–k). Previous studies have linked *fshr-1* to defense responses against pathogens[33], and implicated the

NLP-40/AEX-2 ligand/receptor pair in regulating the defecation cycle[47], suggesting that inefficient removal of pathogens from the intestine may underlie the increased sensitivity of *aex-2* and *nlp-40* mutants. The Golgi pH regulators, GPHR-1 and GPHR-2, acted redundantly to modulate the response to *V. cholerae* (Supplementary Fig. 3k). The enhanced susceptibility of double mutants might be caused by impaired Golgi luminal acidification, thereby disrupting the efficient transport of proteins crucial for defense against infection. The exclusive presence of defects in the double mutants further bolsters the effectiveness of our resource in identifying genes with overlapping functions.

Our screen also isolated several strains that were resistant to *V. cholerae* (Fig. 3e; Supplementary Fig. 3a, b; Supplementary Data 4c). The strain CHS1103, in which *fmi-1* gene was mutated, displayed increased tolerance to *V. cholerae* even though these animals spent extended time on the pathogen (Fig. 3b, e; Supplementary Fig. 3a). The phenotypes were confirmed using *fmi-1* single mutants (Fig. 3i; Supplementary Fig. 3l). *fmi-1* encodes an ortholog of human CELSR2 (cadherin EGF LAG seven-pass G-type receptor 2). In *C. elegans*, FMI-1 is required for neuronal development[48-50], likely playing a role in neuronal regulation of behavioral and innate immune response to the infection. The identification of *nmur-1*, *capa-1*, *frpr-10*, *dop-4*, and *npr-8* confirmed the earlier observations that they are involved in response to bacterial pathogens[31,35,37,39] (Fig. 3h; Supplementary Fig. 3m-q). The involvement of FRPR-9 in the response to bacterial pathogens was supported by the observation that the strain defective in *flp-19*, encoding the putative ligand for FRPR-9, also exhibited increased resistance to *V. cholerae* (Supplementary Fig. 3o, r). *frpr-9* mutants exhibited normal pharyngeal-pumping, pathogen avoidance, and pathogen accumulation in the intestine (Supplementary Fig. 3s–v). Therefore, their increased tolerance to *V. cholerae* was unlikely caused by reduced pathogen uptake or diminished intestinal pathogen accumulation. *frpr-9* mutants were not only resistant to gram-negative bacterial pathogens *V. cholerae* and *P. aeruginosa*, but also exhibited enhanced tolerance to the gram-positive bacterium *E. faecalis* (Supplementary Fig. 3r, w, x), implying that FRPR-9 signaling likely modulates innate immunity in *C. elegans*. Taken together, our study confirmed the roles of previously identified molecules and also uncovered new regulators in behavioral and innate immune responses to the infection.

## Peptidergic signaling modulates AWC-mediated chemotaxis

The olfactory neurons AWA and AWC are the main sensors of volatile attractants in *C. elegans*[14,51], while responses to volatile repellents involve multiple neurons, including ASH, ADL and AWB[51-53]. To date, only a handful of chemoreceptor/ligand pairs have been established[12]. We tested if our mutant collection offered a feasible approach to facilitate the identification of the receptors for various odorants. To this end, we assayed all the mutants in our libraries for their responses to a panel of chemoattractants and repellents (Supplementary Data 5a). The assays were performed as previously described (Fig. 4a)[53]. No chemoreceptor mutant exhibited severe response defects to undiluted odorants (Fig. 4b, c), in line with the notion that high odorant concentrations typically engage multiple receptors of low affinity[51-54].

Screens using the diluted odorants identified three peptidergic mutants (CHS1025, CHS10063 and CHS10013) that exhibited significantly reduced chemotaxis to all AWC− but not AWA− sensed odorants (Fig. 4d, e; Supplementary Data 5a), suggesting that peptidergic signaling modulates the AWC circuit[55]. Inspecting the overall responses to the odorants as well as to the pathogen further highlighted the correlation between CHS1025 and CHS10063 (Fig. 4f; Supplementary Data 5b). The strain CHS1025 disrupted the neuropeptide receptor encoding gene *frpr-9*, whereas CHS10063 contained a mutation in the neuropeptide encoding gene *flp-19* (Fig. 4e). These

observations suggest that FLP-19 may act through FRPR-9 to regulate AWC-mediated chemosensation.

We used low concentrations of 2,3-pentanedione and a different assay strategy to further dissect chemotaxis in these two strains (Supplementary Fig. 4a)[56]. We confirmed that the response to 2,3-pentanedione required FRPR-9 and FLP-19 as well as the neuropeptide FLP-20 (Fig. 5a–c; Supplementary Fig. 4b, c). Both *frpr-9* mutants and *flp-19; flp-20* double mutants showed specific defects in chemotaxis to AWC−sensed odorants (Supplementary Fig. 4d, e). *frpr-9* mutants displayed a reduction similar to that of *flp-19* mutants (Fig. 5c). The defect of *frpr-9; flp-19* double mutants was not significantly different from either single mutant, supporting that FLP-19 likely acts through FRPR-9 to regulate AWC-mediated chemotaxis (Fig. 5c). Consistently, disrupting *frpr-9* did not exacerbate the defects observed in *flp-19; flp-20* double mutants (Fig. 5c). The disruption of *flp-20* alone did not cause any defects in chemotaxis to 2,3-pentanedione (Fig. 5c; Supplementary Fig. 4c), but a synergistic effect was observed when *flp-20* was co-disrupted with *flp-19* (Fig. 5c; Supplementary Fig. 4c). Furthermore, mutating *flp-20* led to a further reduction of chemotaxis in *frpr-9; flp-19* double mutants (Fig. 5c). All these observations imply that FLP-20 likely acts in parallel to FRPR-9/FLP-19 but plays a minor role in chemotaxis, and it is only required for a fraction of the residual responses in animals deficient in FRPR-9/FLP-19 signaling (Fig. 5c). This was further supported by the observations that the defect of *flp-19; flp-20* double mutants was fully rescued by overexpressing *flp-19* genomic DNA under its endogenous promoter (Fig. 5d), but only marginally rescued by expressing *flp-20* genomic DNA under its own promoter (Fig. 5e).

The chemotaxis defect of *frpr-9* mutants was rescued by expressing *frpr-9* genomic DNA under its endogenous promoter (Fig. 5f). We next explored the cellular focus of FRPR-9 and FLP-19 action. By testing a collection of promoters, we found that *flp-19* expression from the promoters *mbr-1* or *sra-11* rescued the chemotaxis defect of *flp-19; flp-20* double mutants (Fig. 5g, h; Supplementary Fig. 5a–g). The expression of these two promoters overlaps in AIN neurons, suggesting that FLP-19 acts in AIN neurons to regulate AWC-mediated odorant responses. FRPR-9 is expressed in neurons including AWC (Supplementary Fig. 5h). Expressing *frpr-9* genomic DNA under various promoters revealed that its expression in AWC neurons was sufficient to rescue the chemotaxis defect of *frpr-9* mutants (Fig. 5i, j; Supplementary Fig. 5i–n), suggesting a peptidergic feedback in the modulation of AWC-mediated olfactory sensation[55]. These data further support the feasibility of our approach in identifying peptidergic signaling modules involved in the regulation of specific cellular processes.

## *srx-1*, *srx-2*, and *srx-3* encode the 2,3-pentanedione receptors

In the screen, we obtained an additional strain CHS1135, displaying markedly decreased attraction to low concentrations of 2,3-pentanedinone (Fig. 4d, e; Supplementary Data 5a). CHS1135 contains mutations in four putative chemoreceptor genes *srx-1*, *srx-2*, *srx-3*, and *srx-4* (Fig. 4e). *srx-1; srx-2; srx-3* triple mutants displayed a reduced chemotaxis that was comparable to that of CHS1135 (Supplementary Fig. 6a), and the defects were specific in response to the diluted 2,3-pentanedinone (Fig. 6a; Supplementary Fig. 6b). Intriguingly, disrupting *srx-2* alone also led to a clear reduction in chemotaxis (Fig. 6b), even though the defect did not reach the level observed in *srx-1; 2; 3* triple mutants (Fig. 6c; Supplementary Fig. 6a). However, *srx-1; 3* double mutants exhibited a normal response (Fig. 6c; Supplementary Fig. 6a). These observations suggest that SRX-2 plays the major role, whereas SRX-1 and SRX-3 contributed to the residual responses when SRX-2 was absent.

Low concentrations of 2,3-pentanedione are sensed by one of the two bilateral AWC neurons, termed AWC$^{OFF}$ neuron[57]. Single-cell analysis indicates that *srx-1, 2* and *3* are all expressed in AWC$^{OFF}$[13]. Consistently, the transcriptional *gfp* reporter driven by *srx-2* promoter was

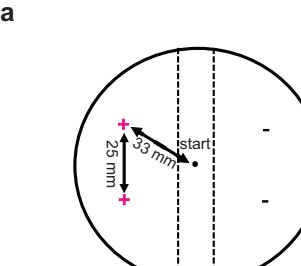

**a**

Chemotaxis index = $\dfrac{(N^+ - N^-)}{(N^+ + N^-)}$

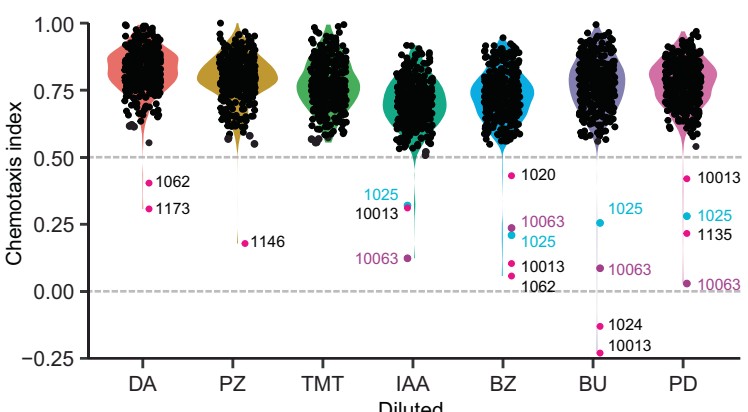

**c**

| Chemical | Strain | Genotype | CI |
|---|---|---|---|
| DA | CHS1131 | lite-1; gur-3; egl-47 | 0.29 |
| | CHS1069 | npr-31; 32; 33 | 0.17 |
| | CHS1103 | fmi-1; lat-2; mth-1; 2 | 0.02 |
| | CHS1018 | y37e11al.1; f56a11.4; b0034.5 | −0.06 |
| | CHS1178 | aex-2; aexr-1; 2; 3 | −0.47 |
| TMT | CHS1083 | ckr-1; 2 | −0.27 |
| | CHS10148 | nlp-11; 54; 66; 79 | −0.27 |
| | CHS10085 | flp-10; 11; 27; 34 | −0.32 |
| | CHS10081 | ins-1; 14; 31; 39 | −0.32 |
| | CHS10103 | nlp-2; 5; 10; 50 | −0.35 |
| | CHS1247 | srh-10; 11; 15; 16; 17; 18; 19; 20; 21; 22; 23 | −0.36 |
| | CHS10006 | flp-7; 9; 17; 22 | −0.38 |
| | CHS1076 | ntr-1; 2 | −0.38 |
| | CHS1262 | srt-42; 43; 44; 45; 47; 48; 49; 50; 52; 53; y57g11c.28; y57g11c.30 | −0.39 |
| | CHS10089 | nlp-36; 70; 76 | −0.41 |
| | CHS1104 | srv-5; 6; 7; 8; 9; 10; c34b4.3; b0563.6 | −0.41 |
| | CHS10109 | nlp-21; 69; 73; pdf-1 | −0.43 |
| | CHS10010 | ins-10; 19; 35; 36 | −0.46 |
| | CHS10118 | ins-20; 24; 30; 33; daf-28 | −0.47 |
| | CHS10034 | flp-3; 6; 13; 18 | −0.47 |
| IAA | CHS10109 | nlp-21; 69; 73; pdf-1 | −0.38 |
| | CHS10149 | nlp-6; 12; 39; capa-1 | −0.44 |
| OCT | CHS1127 | dmsr-4; 5; 6; 7; 8 | −0.38 |

**e**

| Chemicals | Neuron | Strain | Genotype | Identified gene(s) | CI |
|---|---|---|---|---|---|
| DA (1:2000) | AWA | CHS1173 | odr-10 | odr-10 | 0,31 |
| | | CHS1062 | npr-6; 7; 8 | Undetermined | 0,4 |
| PZ (1:1000) | AWA | CHS1146 | srx-48; 50; 51; 54; 59; 64 | srx-64 | 0,18 |
| IAA (1:200) | AWC (AWA) | CHS10063 | flp-19; 20; 26; 32; 33 | flp-19, flp-20 | 0,12 |
| | | CHS10013 | nlp-8; 16; 55; 61 | Undetermined | 0,31 |
| | | CHS1025 | frpr-7; 8; 9; 10 | frpr-9 | 0,32 |
| BZ (1:1000) | AWC (AWA) | CHS1062 | npr-6; 7; 8 | Undetermined | 0,06 |
| | | CHS10013 | nlp-8; 16; 55; 61 | Undetermined | 0,1 |
| | | CHS1025 | frpr-7; 8; 9; 10 | frpr-9 | 0,21 |
| | | CHS10063 | flp-19; 20; 26; 32; 33 | flp-19, flp-20 | 0,24 |
| | | CHS1020 | r13h7.2; k03h6.5; k03h6.1; t10e10.3 | Undetermined | 0,43 |
| BU (1:10000) | AWC^ON | CHS10013 | nlp-8; 16; 55; 61 | Undetermined | −0,23 |
| | | CHS1024 | pdfr-1; seb-2; 3 | Undetermined | −0,13 |
| | | CHS10063 | flp-19; 20; 26; 32; 33 | flp-19, flp-20 | 0,09 |
| | | CHS1025 | frpr-7; 8; 9; 10 | frpr-9 | 0,26 |
| PD (1:10000) | AWC^OFF | CHS10063 | flp-19; 20; 26; 32; 33 | flp-19, flp-20 | 0,03 |
| | | CHS1135 | srx-1; 2; 3; 4 | srx-1, srx-2, srx-3 | 0,22 |
| | | CHS1025 | frpr-7; 8; 9; 10 | frpr-9 | 0,28 |
| | | CHS10013 | nlp-8; 16; 55; 61 | Undetermined | 0,42 |

**f**

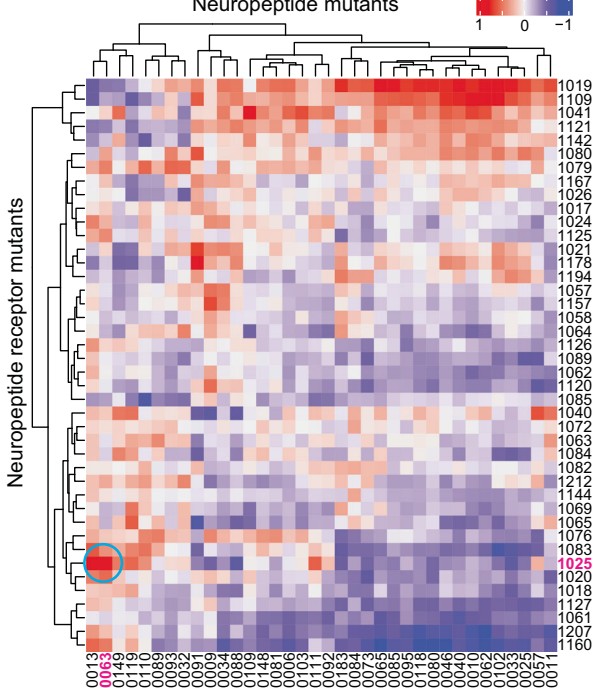

expressed in a limited number of neurons, including AWC^OFF (Fig. 6d). Moreover, an endogenously tagged SRX-2::GFP fusion protein was localized to the cilia of AWC (Supplementary Fig. 6c). Expressing *srx-2* genomic DNA under its endogenous promoter or selectively in AWC^OFF neuron fully restored the chemotaxis to *srx-2* mutants (Fig. 6e, f). Overexpressing *srx-1* and *srx-3* in AWC^OFF neuron also rescued the

defect of *srx-2* mutants (Fig. 6g), further confirming the related functions of these proteins.

To further validate a role for SRX-2 in mediating responses to low concentrations of 2,3-pentanedione, we assessed odorant-evoked intracellular calcium dynamics in AWC^OFF neurons. The transgenic strain expressing the genetically encoded Ca²⁺ sensor GCaMP3 in AWC

**Fig. 4 | The identification of chemoreceptor and peptidergic mutants that are defective in response to volatile odorants. a** Plate format of population chemotaxis assays used in the screen. **b** Violin plots of chemotaxis indices of GPCR and neuropeptide mutants to undiluted diacetyl (DA), 2, 4, 5-trimethylthiazole (TMT), isoamyl alcohol (IAA), benzaldehyde (BZ), 2,3-pentanedione (PD), 2-nonanone (NON) and 1-octanone (OCT). The strains with chemotaxis indices (>−0.5) were indicated in magenta. **c** The genotypes of mutants that were indicated in (**b**). CI refers to chemotaxis index. **d** Violin plots of chemotaxis indices of GPCR and neuropeptide mutants to 1:2000 diacetyl (DA), 1:1000 pyrazine (PZ), 1:2000 2, 4, 5-trimethylthiazole (TMT), 1:200 isoamyl alcohol (IAA), 1:1000 benzaldehyde (BZ),

1:10,000 2-butanone (BU), and 1:10,000 2,3-pentanedione (PD). The strains that were defective in chemotaxis to each odorant were indicated. **e** The genotypes of mutants that were defective in chemotaxis to various odorants as indicated in (**d**). The relevant genes in the strains CHS1025, CHS1135, CHS1146, CHS1173 and CHS10063 have been identified and listed in the table. CI refers to chemotaxis index. **f** Heatmap of the correlation between neuropeptide and neuropeptide receptor mutants in response to both odorants and *V. cholerae*. Red or blue color indicates the positive or negative correlation, respectively. The neuropeptide receptor mutant CHS1025 and the neuropeptide mutant CHS10063 are indicated in magenta, and their correlation is highlighted within a cyan circle.

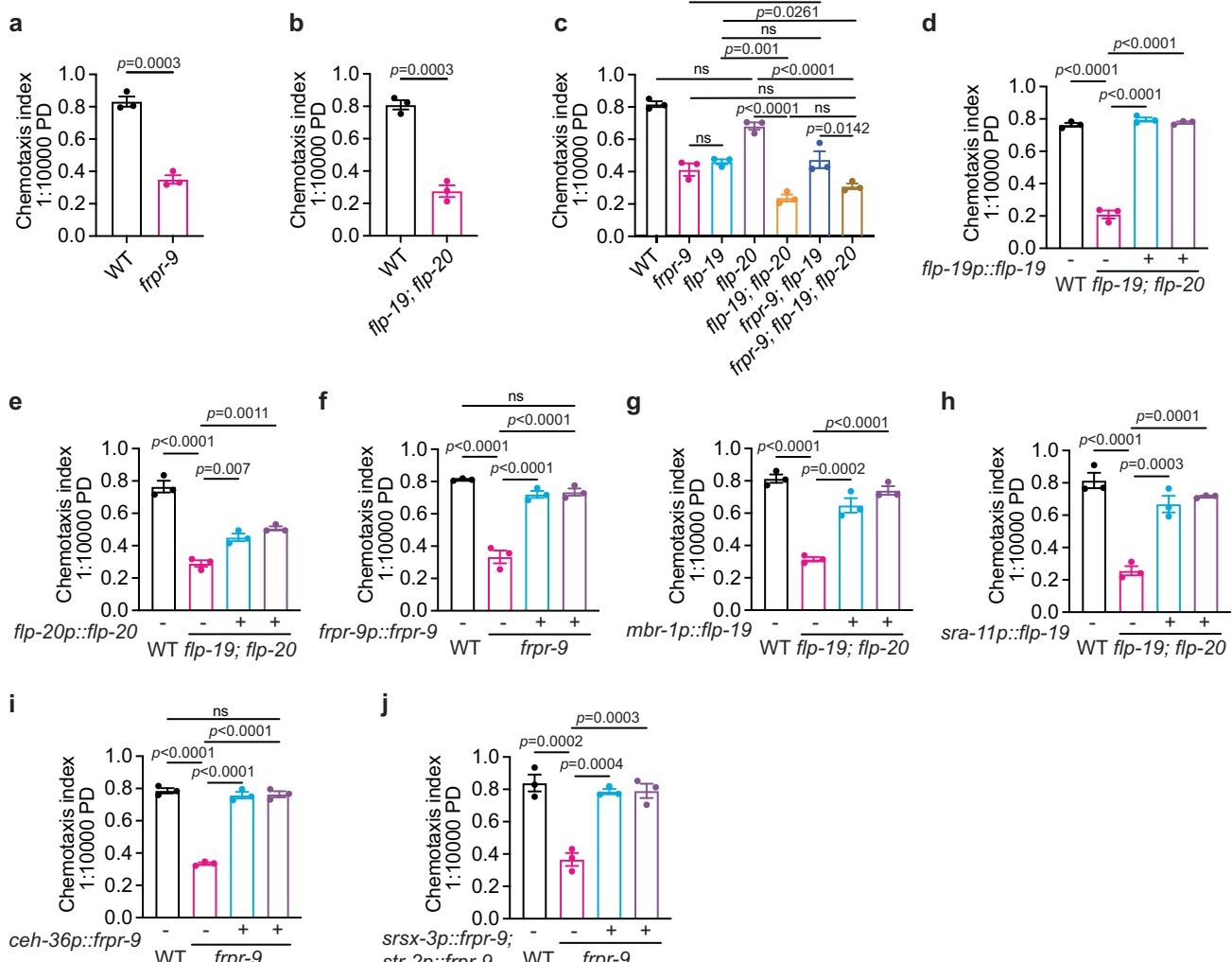

**Fig. 5 | Neuropeptide FLP-19 acts on FRPR-9 receptor to regulate AWC mediated chemotaxis. a–c** Chemotaxis indices to 1:10000 diluted 2,3-pentanedione (PD) of animals with indicated genotypes WT and *frpr-9(yum1004)* in (**a**), WT and *flp-19(yum1005); flp-20(yum1006)* double mutants in (**b**), and WT, *frpr-9(yum1004)*, *flp-19(yum1005)*, *flp-20(yum1062)*, *flp-19(yum1005); flp-20(yum1006)* double, *frpr-9(yum1004); flp-19(yum1005)* double and *frpr-9(yum1004); flp-19(yum1005) flp-20(yum1006)* triple mutants in (**c**). *p* values are displayed in the plot. Two-tailed *t* test for (**a**) and (**b**), and one-way ANOVA, Tukey's multiple comparison for (**c**). **d, e** Chemotaxis indices to 1:10000 diluted 2,3-pentanedione (PD) in WT, *flp-19(yum1005); flp-20(yum1006)* and two independent lines of transgenic *flp-19(yum1005); flp-20(yum1006)* expressing *flp-19* genomic DNA from its endogenous promoter in (**d**), or expressing *flp-20* genomic DNA from its endogenous promoter in (**e**). *p* values are displayed in the plot. One-way ANOVA, Tukey's multiple comparison. **f** Chemotaxis indices to 1:10000 diluted 2,3-pentanedione (PD) of animals with indicated genotypes WT, *frpr-9(yum1004)* and two independent lines of

transgenic *frpr-9(yum1004)* expressing *frpr-9* genomic DNA from its endogenous promoter. *p* values are displayed in the plot. One-way ANOVA, Tukey's multiple comparison. **g, h** Chemotaxis indices to 1:10000 diluted 2,3-pentanedione (PD) of animals with indicated genotypes WT (N2), *flp-19(yum1005); flp-20(yum1006)* and two independent lines of transgenic *flp-19(yum1005); flp-20(yum1006)* expressing *flp-19* cDNA from *mbr-1* promoter in (**g**), or from *sra-11* promoter in (**h**). *p* values are displayed in the plot. One-way ANOVA, Tukey's multiple comparison. **i, j** Chemotaxis indices to 1:10000 diluted 2,3-pentanedione (PD) of animals with indicated genotypes WT (N2), *frpr-9(yum1004)* and two independent lines of transgenic *frpr-9(yum1004)* expressing *frpr-9* genomic DNA in AWC neurons from *ceh-36* promoter in (**i**), or from both *srsx-3* and *str-2* promoters in (**j**). *p* values are displayed in the plot. One-way ANOVA, Tukey's multiple comparison. In all the figure panels (**a–j**), data were generated from *n* = 3 biological replicates, and are presented as mean values +/- SEM. Source data are provided as a Source Data file.

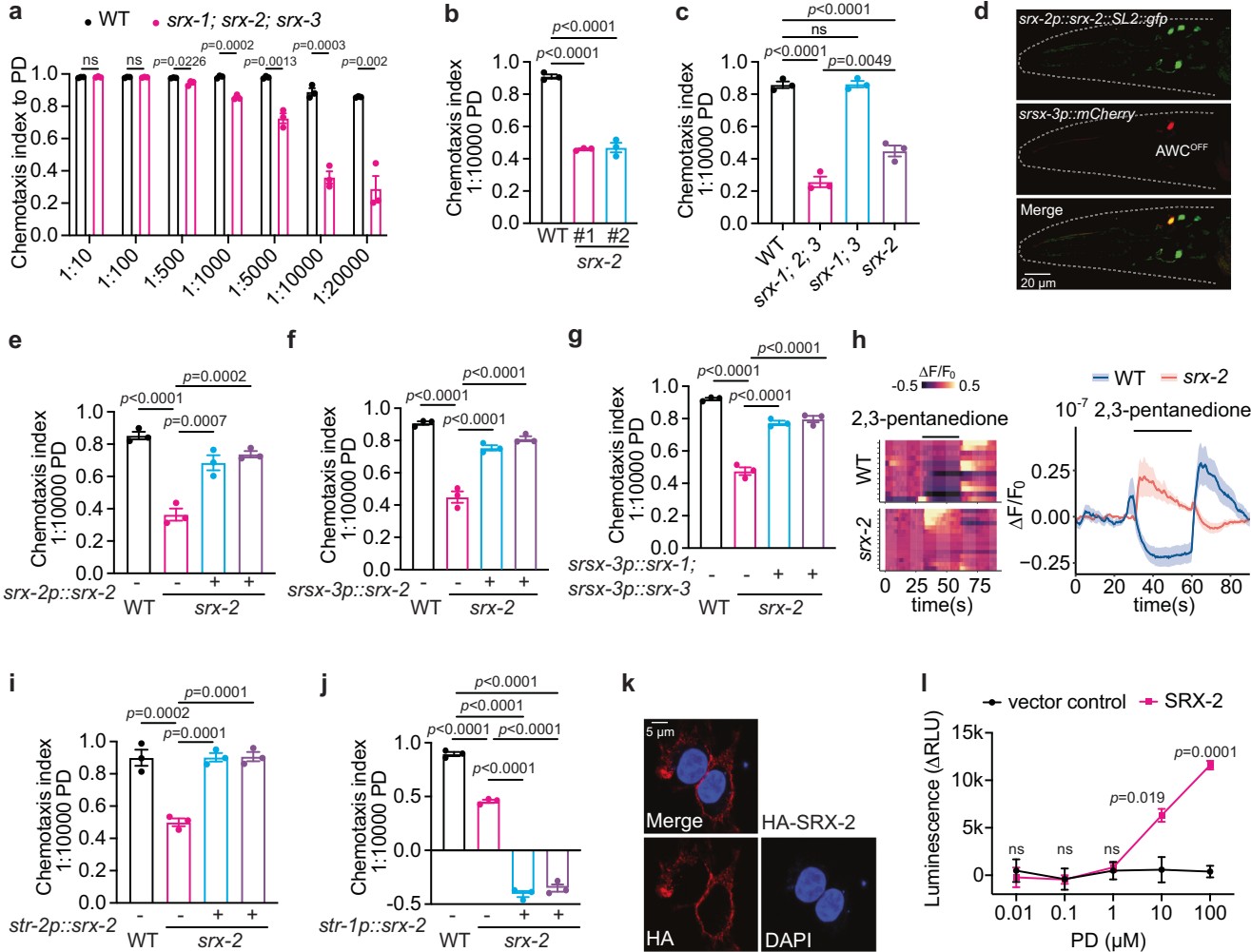

**Fig. 6 | SRX-1, SRX-2 and SRX-3 are the receptors for 2,3-pentanedione.**
**a** Chemotaxis indices to different dilutions of 2,3-pentanedione (PD) in animals with indicated genotypes. *p* values are displayed in the plot. Two-tailed *t* test. **b, c** Chemotaxis indices to 1:10000 diluted 2,3-pentanedione (PD) in animals with indicated genotypes. #1 and #2 in (**b**) indicate two independent null alleles of *srx-2*. *p* values are displayed in the plot. One-way ANOVA, Tukey's multiple comparison. **d** GFP is expressed from a *srx-2p::srx-2::SL2::gfp* polycistronic construct. mCherry expression under *srsx-3* promoter indicates AWC^OFF neuron. **e–g** Chemotaxis indices to 1:10000 diluted 2,3-pentanedione (PD) in WT, *srx-2(yum1007)* and two independent lines of transgenic *srx-2(yum1007)* expressing *srx-2* genomic DNA from its endogenous promoter (**e**), expressing *srx-2* cDNA from *srsx-3* promoter in AWC^OFF neuron (**f**), or simultaneously expressing *srx-1* and *srx-3* cDNAs from *srsx-3* promoter in AWC^OFF neuron (**g**). *p* values are displayed in the plot. One-way ANOVA,

Tukey's multiple comparison. **h** Heatmap (left) and average values (right) of GCaMP3 fluorescence intensity changes in response to $10^{-7}$ diluted 2,3-pentanedione in AWC^OFF neuron of WT and *srx-2(yum1007)*. **i, j** Chemotaxis indices to 1:10000 diluted 2,3-pentanedione (PD) in WT, *srx-2(yum1007)* and two independent lines of transgenic *srx-2(yum1007)* expressing *srx-2* cDNA from *str-2* promoter in AWC^ON neuron (**i**), or from *str-1* promoter in AWB neurons (**j**). *p* values are displayed in the plot. One-way ANOVA, Tukey's multiple comparison. **k** Cell surface expression of HA-SRX-2 stained with anti-HA antibody (red) and nuclei stained with DAPI (blue). **l** Intracellular cAMP concentrations in response to different dilutions of 2,3-pentanedione in SRX-2 (magenta) and vector (black) transfected cells. *p* values are displayed in the plot. Two-tailed *t* test. In all figure panels (**a–c, e–g, i, j**, and **l**), data were generated from *n* = 3 biological replicates, and are presented as mean values +/- SEM. Source data are provided as a Source Data file.

neurons displayed normal chemotaxis to 2,3-pentanedione (Supplementary Fig. 6d). 2,3-pentanedione elicits a robust reduction of intracellular calcium levels in AWC^OFF neurons of wild type animals[58], which was abolished in *srx-2* mutants with a small fraction of mutants instead showing a small increase of intracellular calcium levels in AWC^OFF neurons (Fig. 6h). Calcium transients evoked by isoamyl alcohol (IAA) in AWC neurons were not affected in *srx-2* mutants (Supplementary Fig. 6e). These observations indicate that SRX-2 is necessary for 2,3-pentanedione-evoked responses in AWC^OFF neuron.

To determine if SRX-2 is sufficient to confer 2,3-pentanedione responses, we ectopically expressed *srx-2* in other neurons of *C. elegans*, or in a heterologous system. When *srx-2* was expressed in AWC^ON neurons in *srx-2* mutants, it restored 2,3-pentanedione chemotaxis to the mutant animals (Fig. 6i). AWB neurons mediate avoidance of

volatile repellents[59]. Mis-expressing the diacetyl receptor ODR-10 in the AWB neurons of *odr-10* null mutants, causes animals to be repelled by normally attractive levels of diacetyl[59]. When *srx-2* was ectopically expressed in AWB neurons in *srx-2* single or *srx-1; 2; 3* triple mutants, animals were repelled by the diluted 2,3-pentanedione (Fig. 6j; Supplementary Fig. 6f). Mis-expressing *srx-1* or *srx-3* in AWB neurons in the triple mutants resulted in a similar, albeit milder, aversion (Supplementary Fig. 6f), implying that SRX-1 and SRX-3 may possess a lower affinity for 2,3-pentanedione or lower efficiency in eliciting behavioral response. This observation was confirmed in single worm chemotaxis assays, showing that the misexpression of *srx-1* in AWB neurons of triple mutants was less effective in triggering an aversive response compared to *srx-2* (Supplementary Fig. 6g). When co-expressing *srx-1* and *srx-2* ectopically in AWB neurons, the repulsive effect closely

resembled that of *srx-2* misexpression alone (Supplementary Fig. 6g), supporting that SRX-2 likely serves as the primary receptor for low concentrations of 2,3-pentanedione. However, it is plausible that each receptor might have a predominant role in a specific context, dependent on the odor concentrations present in the environment. The presence of three receptors with different expression levels and different affinities might also increase the efficiency and fidelity for the response to varying concentrations of 2,3-pentanedione.

We also sought to express *srx-2* in a heterologous system. cAMP assays have been widely used to monitor the activation of odorant receptors expressed in heterologous cell lines such as HEK293 and HEK293T-derived Hana3A cells[60–63]. Odorant binding to the olfactory receptor induces conformational change in the receptor, which in turn binds and activates Gαs, leading to an increased production of intracellular cAMP. We tagged SRX-2 with a HA epitope at its N-terminus. This chimeric protein was successfully targeted to the plasma membrane of HEK293T cells as indicated by cell surface staining (Fig. 6k). Addition of 2,3-pentanedione, but not isoamyl alcohol, activated SRX-2 as indicated by an increase of cAMP in SRX-2 transfected HEK293T cells but not in control cells transfected with vector alone (Fig. 6l; Supplementary Fig. 6h). These observations indicate that SRX-2 is the cognate receptor for 2,3-pentanedione, and demonstrate the feasibility of our approach to deorphanize odorant receptors.

### *srx-64* encodes a pyrazine receptor

We found that the strain CHS1146 exhibited defects in chemotaxis to low concentrations of the volatile odorant pyrazine (Fig. 4d). This strain contains mutations in six putative chemoreceptor genes *srx-48, srx-50, srx-51, srx-54, srx-59*, and *srx-64* (Fig. 4d, e; Supplementary Data 5a). Disrupting *srx-64* alone led to a behavioral phenotype similar to that of the original strain (Supplementary Fig. 7a). This observation was confirmed using a second null allele of *srx-64* (Fig. 7a). The defect of *srx-64* mutants was fully complemented by expressing *srx-64* genomic DNA from its endogenous promoter (Fig. 7b). *srx-64* mutants displayed less pronounced defects at high concentrations of pyrazine (Fig. 7c), and responded normally to other volatile attractants as well as to a panel of repellents (Fig. 7d).

Pyrazine is sensed by the AWA neurons[51]. GFP signal from a transcriptional reporter using *srx-64* promoter was exclusively observed in AWA (Fig. 7e), and the GFP-tagged SRX-64 was localized to the cilia of these neurons (Fig. 7f). Expressing *srx-64* in AWA neurons using *odr-10* promoter restored pyrazine chemotaxis to *srx-64* mutants (Fig. 7g). These observations suggest that SRX-64 acts in AWA neurons. We next sought to monitor pyrazine-evoked calcium transients in AWA neurons[64–66]. The transgenic animals expressing the genetically encoded $Ca^{2+}$ sensor GCaMP2 specifically in AWA showed normal chemotaxis to the diluted pyrazine (Supplementary Fig. 7b). In wild type animals, pyrazine triggered a rapid increase of intracellular calcium in AWA neurons (Fig. 7h), which was eliminated in *srx-64* mutants (Fig. 7h). However, diacetyl-evoked calcium changes in AWA neurons were unaffected (Supplementary Fig. 7c). Furthermore, mis-expressing *srx-64* in AWB neurons in *srx-64* mutants was sufficient to trigger aversion to low concentrations of pyrazine (Fig. 7i). These results indicate that SRX-64 is likely a pyrazine receptor.

To further explore the sufficiency of SRX-64 to mediate pyrazine responses, we expressed *srx-64* in a heterologous system, following a similar procedure that was used for SRX-2. HA-tagged SRX-64 was successfully inserted to the plasma membrane of HEK293T cells (Fig. 7j). Application of pyrazine stimulated the production of intracellular cAMP in SRX-64 transfected cells, but not in the vector-transfected cells (Fig. 7k). In contrast, diacetyl administration failed to induce noticeable rise of cAMP levels in either SRX-64 or vector-transfected cells (Supplementary Fig. 7d). Taken together, these observations indicate that SRX-64 is the cognate receptor for low concentrations of pyrazine.

## Discussion

Although systematic efforts have led to the deorphanization of many neuropeptide receptors in *C. elegans*[15], the majority of GPCRs, particularly chemoreceptors, have not yet been associated with ligands. This reflects the very large number of chemoreceptors and the likelihood of functional redundancy. Novel strategies, which allow for the comprehensive dissection of GPCR signaling, are needed to complement existing approaches such as forward genetic screens, biochemistry, candidate gene approaches, computational modeling, and expression analyses. To this end, we disrupted 1654 GPCR genes encoded by the *C. elegans* genome in 284 strains, covering nearly all GPCR encoding genes annotated in Wormbase. We show that this mutant collection can be used to identify GPCRs involved in the cellular processes of interest. The limited number of strains makes the screens of a manageable scale. Unlike forward genetic screens that are often constrained by extensive mapping to find the causal mutations, the known genotypes in each strain allow for the rapid identification of the relevant GPCRs. In parallel, we disrupted all neuropeptide genes in 38 strains. The combination of two mutant libraries enables us to trace GPCR-mediated flow of information from the sensory inputs to the internal cell-to-cell communication.

To demonstrate the value of our resource, we performed three screens to identify GPCRs involved in acute hypoxia sensation, responses to pathogen, and olfactory sensing. The successful isolation of relevant GPCRs and neuropeptides in each process demonstrates the feasibility of our approach to dissect GPCR signaling in *C. elegans*. Our GPCR mutant library enables the simultaneous and unbiased assessment of nearly all GPCRs, significantly enhancing the likelihood of identifying the relevant receptors in a specific cellular process. In addition, screening the mutants in our collections markedly increases the probability of discovering genes with redundant functions. For example, it is unlikely that three receptors, which act redundantly to modulate acute hypoxia response, would be identified by alternative approaches. Furthermore, the identification of the long-sought receptors for pyrazine and 2,3-pentanedione illustrates the potential of our mutant collection in expediting the discovery of putative receptors for a variety of small molecules.

We note, however, that a small set of GPCRs are still missing in our library, and our approach to cluster GPCR genes into individual groups is not optimal, which leads to gene redundancy not being fully eliminated. To overcome these, our on-going efforts are focused on disrupting the remaining GPCR-encoding genes and mutating multiple receptor genes that are expressed in the same neurons as indicated by the gene expression profiling[13]. It is also possible that optimization of assay conditions may be necessary in specific screens to effectively identify the potential odorant receptors. For example, our data showed improved consistency and reliability when we modified the assay conditions to examine the responses of *frpr-9* and *flp-19* mutants to 2,3-pentanedione, as compared to the conditions used in the initial screen (Supplementary Fig. 4a)[56]. Moreover, specific odorants may be sensed by both broadly and narrowly tuned GPCRs and neurons, making it more difficult to identify specific receptors in the absence of a sensitized background. While acknowledging current limitations, we are confident that our resource offers an excellent starting point to dissect GPCR-mediated sensory perception and signal transduction in *C. elegans*, and provides a feasible and high throughput approach to tackle many challenging questions related to GPCR biology in this organism.

## Methods

### Strains

*C. elegans* was maintained under standard laboratory conditions[67]. The GPCR and neuropeptide mutant libraries, and other strains used in this study are provided in Supplementary Data 1, 2 and 6. Strains in the mutant collections usually contain multiple mutations. To identify the

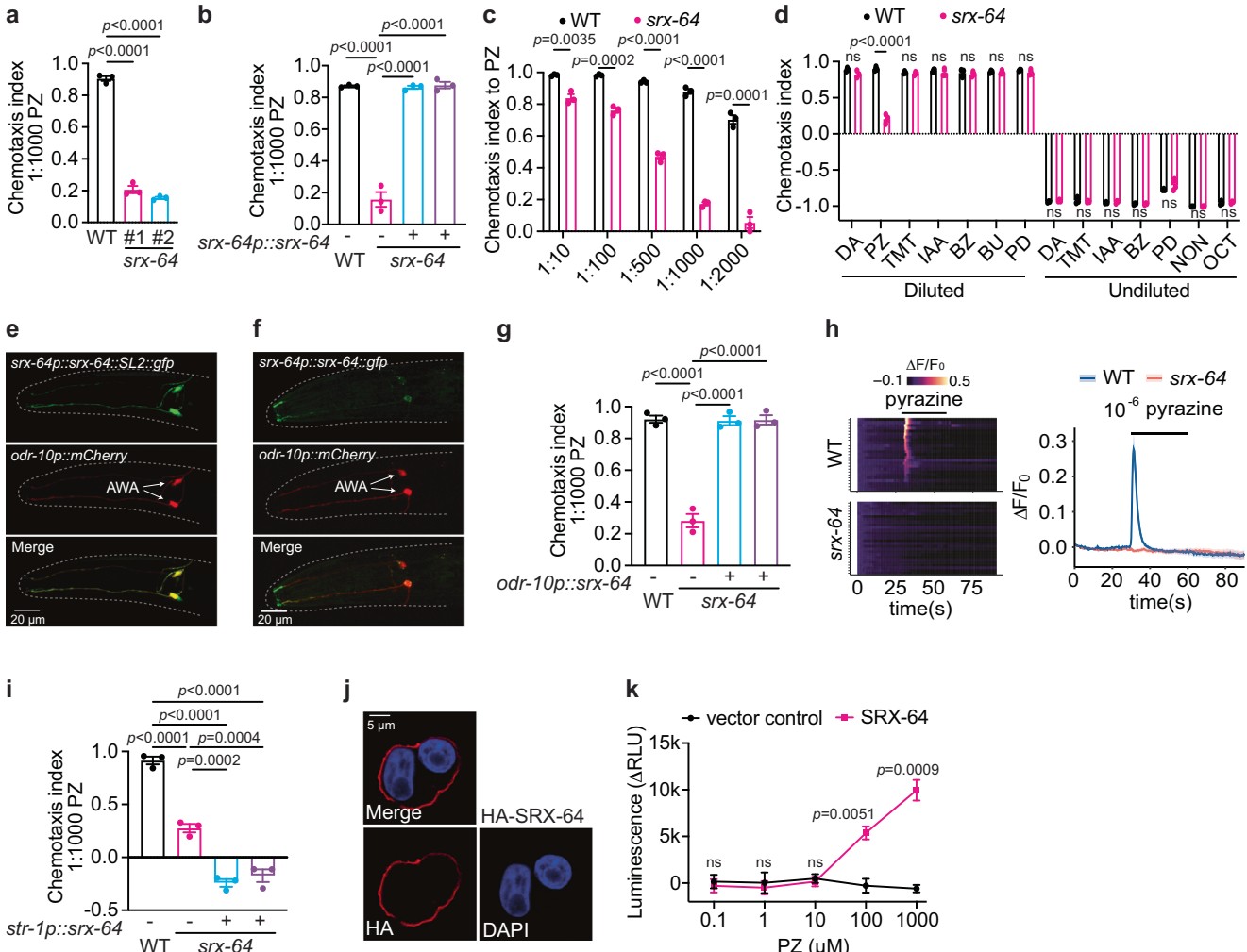

**Fig. 7 | SRX-64 is a cognate receptor for the odorant pyrazine. a, b** Chemotaxis indices to 1:1000 diluted pyrazine (PZ) of animals with indicated genotypes. #1 and #2 indicate two independent null alleles of *srx-64* in (**a**). *srx-64* genomic DNA was expressed from its endogenous promoter in *srx-64(yum1002)* in (**b**). *p* values are displayed in the plot. One-way ANOVA, Tukey's multiple comparison. **c** Chemotaxis indices to different dilutions of pyrazine (PZ) in WT and *srx-64(yum1002)*. *p* values are displayed in the plot. Two-tailed *t* test. **d** Chemotaxis indices of WT and *srx-64(yum1002)* to various diluted and undiluted odorants. *p* values are displayed in the plot. Two-tailed *t* test. **e** GFP is expressed from a *srx-64p::srx-64::SL2::gfp* poly-cistronic construct. mCherry expression under *odr-10* promoter indicates AWA neurons. **f** GFP is expressed from a *srx-64p::srx-64::gfp* construct. mCherry under *odr-10* promoter indicates AWA neurons. **g** Chemotaxis indices to 1:1000 diluted pyrazine (PZ) of animals with indicated genotypes. *srx-64* cDNA was expressed in AWA neurons under *odr-10* promoter in *srx-64(yum1002)*. *p* values are displayed in

the plot. One-way ANOVA, Tukey's multiple comparison. **h** Heatmap (left) and average values (right) of GCaMP2 fluorescence intensity changes to $10^{-6}$ diluted pyrazine in AWA neurons of WT and *srx-64(yum1002)*. **i** Chemotaxis indices to 1:1000 diluted pyrazine (PZ) of animals with indicated genotypes. *srx-64* cDNA was expressed in AWB neurons under *str-1* promoter in *srx-64(yum1002)*. *p* values are displayed in the plot. One-way ANOVA, Tukey's multiple comparison. **j** Cell surface expression of HA-SRX-64 stained with anti-HA antibody (red) and nuclei stained with DAPI (blue) in HEK293T cells. **k** Intracellular cAMP concentrations in response to different dilutions of pyrazine in SRX-64 (magenta) or vector (black) transfected HEK293T cells. *p* values are displayed in the plot. Two-tailed *t* test. In all figure panels (**a–d, g, i, and k**), data were generated from *n* = 3 biological replicates, and are presented as mean values +/- SEM. Source data are provided in the Source Data file.

causal mutations responsible for the phenotypes, the original strains were typically crossed with wild type (N2), and the offspring in subsequent generations were genotyped to obtain a set of strains with different combinations of mutations. These mutants were assessed for their responses to the relevant stimuli. If the mutations are genetically linked, new alleles of these genes were generated using CRISPR/Cas9, either independently or in combination. The resulting strains were subject to phenotypic analyses.

## Molecular Biology
The expression vectors used in *C. elegans* were constructed using Multisite Gateway System (ThermoFisher Scientific), and were listed in Supplementary Data 7a. The promoters, including *dsmr-8* (2.2 kb), *dmsr-7* (1.8 kb), *flp-1* (0.5 kb), *srx-64* (1.8 kb), *odr-10* (1.1 kb), *str-1* (4 kb),

*str-2* (3.7 kb), *srx-2* (2.6 kb), *srsx-3* (0.9 kb), *odr-3* (2.7 kb), *odr-1* (2.4 kb), *frpr-9* (2.9 kb), *flp-19* (2.8 kb), *flp-20* (2.8 kb), *mbr-1* (5.3 kb), *sra-11* (4.4 kb), *ceh-36* (0.37 kb), *mgl-1* (0.23 kb), *gcy-28.d* (2.9 kb), *flp-18* (3.6 kb), *glr-1* (5.3 kb), *nmr-1* (2.2 kb), *tph-1* (2 kb), *odr-2.b* (2.5 kb), *osm-6* (2.7 kb), *gpa-3* (6 kb), *flp-5* (2.3 kb), were amplified from genomic DNA and assembled to pDONR P4P1 using BP reaction. Genes of interest, including cDNAs (*srx-64, srx-1, srx-2, srx-3, dmsr-8, flp-1* and *flp-19*) and genomic sequences (*dmsr-7, srx-64, srx-2, frpr-9, flp-19* and *flp-20*) were amplified using either the first strand cDNA library or genomic DNA as the template and cloned into pDONR 221 with BP clonase. The expression plasmids were assembled using LR reaction. To express *srx-2* and *srx-64* in HEK293T cells, the cDNAs of *srx-2* and *srx-64* were PCR-amplified, digested with BamHI and XhoI (*srx-2*) or with BamHI and XbaI (*srx-64*), and cloned onto pcDNA3, which contained HA epitope

sequence for tagging at N terminus. The primers used for cloning were listed in Supplementary Data 7b.

To generate transgenic animals, the expression constructs for *srx-64* and *srx-2* genes were injected at 5 ng/ul together with 50 ng/µl coelomocyte co-injection marker and 50 ng/µl 1 kb DNA ladder. All other constructs were injected at 50 ng/ul together with 50 ng/µl coelomocyte marker.

## In search of GPCR-encoding genes
Sequential steps were undertaken to obtain the list of GPCR-encoding genes. First, the full list of annotated GPCR-encoding genes within each sub-family were obtained by searching them under the directory of 'gene class' using GPCR subfamily names (e.g., *sra*) on Wormbase homepage. Second, the paralogs of each GPCR gene were obtained from gene information page in Wormbase. These paralogs were evaluated using GPCRHMM (https://gpcrhmm.sbc.su.se/), and the number of transmembrane domains were analyzed using TMHMM (https://services.healthtech.dtu.dk/services/TMHMM-2.0/). Third, the protein sequence of each GPCR was used as a template to BLAST search for similar proteins. The hits were analyzed using GPCRHMM and TMHMM, and the putative GPCRs were kept for further analyses.

## Clustering of GPCR genes into subgroups
The main criterion for clustering GPCR-encoding genes is their protein sequence identity. All protein sequences were uploaded to the CD-HIT server (https://sites.google.com/view/cd-hit)[21]. A cut-off of >40% protein sequence identity was used to cluster GPCRs into individual groups, which resulted in groups of varying sizes ranging from 1 to 21 genes. The remaining genes were further clustered into specific groups, based on their genetic positions and phylogenetic relationship with existing genes within the group. The rest of genes were arbitrarily assigned to specific groups according to the numerical order of their gene names. However, we took the following factors into consideration. We aimed to minimize the number of genes disrupted in each strain. First, it was uncertain if multiple rounds of CRISPR/Cas9 genome editing would significantly increase nonspecific mutations or cause genome rearrangement in the background. Second, multiple rounds of genome editing in the same strain were time-consuming. To ensure time efficiency, most strains were subject to a maximum of 2 rounds of editing (≤6 genes) (Supplementary Data 1e). Third, GPCRs from different sub-families were not disrupted in the same strain. For example, *sro-1* was the only gene edited in the strain CHS1013 (Supplementary Data 1e). Moreover, genes with well-established functions were often kept separate from others (Supplementary Data 1e).

## CRISPR/Cas9 genome editing
Genome editing was performed using published protocols[19,20], which employed homology-directed integration of single-strand DNA oligo (ssODN) to repair double-strand DNA breaks. The ssODN contains two 35-base homology arms that flanked the targeted PAM sites. The integration of ssODN led to the insertion of stop codons and a unique restriction enzyme cutting site for genotyping. Additionally, a short coding sequence was removed to generate frameshift.

Up to 3 genes were targeted in each microinjection. The editing efficiency was decreased when 4 genes were targeted simultaneously. In cases where disrupting more than 3 genes in a single strain was necessary, multiple rounds of injections were required to sequentially knock-out all the genes. Repetitive editing attempts in the same strain did not have a clear effect on editing efficiency. When one gene was targeted, 0.5 µl Cas9 (10 µg/µl, IDT, #1081059), 5 µl tracrRNA (0.4 µg/µl in IDT duplex buffer, IDT, #1072534), and 2.8 µl crRNA (0.4 µg/µl in IDTE pH7.5) were mixed and incubated at 37 °C for 10 minutes. Then, 2.2 µl ssODN (synthesized by IDT) (1 µg/µl in nuclease free $H_2O$), 2 µl rol-6 co-injection marker (600 ng/µl in nuclease-free $H_2O$) and 7.5 µl

nuclease-free $H_2O$ were added to form the complete injection mixture. The injection mixture was centrifuged at 18 000x *g* for 10 minutes at room temperature, and 17 µl was transferred to a new tube for injection. The same procedure was used to prepare the injection mixture for disrupting multiple genes, with the quantities of each component adjusted accordingly. To disrupt two genes simultaneously, the mixture includes 0.5 µl Cas9, 6 µl tracrRNA, 2 µl of each crRNA, 2.5 µl of each ssODN, 2 µl rol-6 marker and 2.5 µl nuclease-free $H_2O$. For the disruption of three genes simultaneously, the injection mixture consists of 0.5 µl Cas9, 6 µl tracrRNA, 1.9 µl of each crRNA, 2.1 µl of each ssODN, and 2 µl rol-6 marker. For four genes, 0.5 µl Cas9, 6 µl tracrRNA, 1.7 µl of each crRNA, 1.9 µl of each ssODN, and 2 µl rol-6 marker are used.

In the initial round, we typically targeted three genes simultaneously. After injections, twenty-four transgenic F1 rollers were singled. Animals in F1 or subsequent generations were lysed to genotype the integration of ssODN at each of three targeting sites. Two genotyping primers flanking each edited site were utilized to amplify the fragments of 400 to 1000 bp. If the ssODN was inserted, the PCR products would be cleaved into two fragments of different sizes by the restriction enzymes, whereas PCR products remained undigested if the gene was unedited. Generally, F1 rollers that were heterozygous mutants for all three targeted sites were retained to obtain homozygous mutants in their subsequent generations. After obtaining the triple homozygous mutants, we proceeded with the next round of micro-injection to disrupt three additional genes in the mutant background. The same procedure was used to genotype the disruption of three new genes. This process was repeated until all genes within the group were disrupted. Upon completion, all edited sites in each strain were validated again using PCR-based genotyping, and the final strains were genome-sequenced (Supplementary Data 1f). Many genotyping primers listed in Supplementary Data 1d only yielded PCR products when longAMP Taq polymerase (NEB, M0323L) was used.

To tag *srx-2* with GFP on chromosome, the insertion template was generated by amplifying GFP sequence with primers that had homology arms flanking *srx-2* stop codon. The PCR product was cleaned using AMPure XP beads (A63880, Beckman Coulter), and the injection mix was prepared as previously described[19,20].

## Phylogenetic tree analysis
The longest protein sequences for each gene were extracted using WormMine. A multiple-sequence alignment was performed. A phylogenetic tree was then computed using VeryFastTree v3.2.1[68]. The tree was plotted using the R package ape.

## Genomic DNA extraction
Animals were grown on 9 cm plates until starvation. 2 plates of worms were collected and washed 3 times in M9 buffer. Genomic DNA was extracted using DNeasy Blood and Tissue kit (Qiagen, 69504). DNA concentration was determined using Qubit (Thermo Fisher Scientific). 1 µg of genomic DNA in the elution buffer was used for library preparation. The whole genome sequencing was performed using Illumina Hiseq 4000 at *SciLifeLab*, Sweden.

## Genome sequencing analysis
Whole-genome sequencing reads were mapped to *C. elegans* reference genome (PRJNA13758, WS235) using BWA MEM *(*version 0.7.17)[69]. Reads were deduplicated using Picard (version 2.19.2) (http://broadinstitute.github.io/picard). To examine if genes were disrupted, we ran the CNV detection tool Pindel (version 0.2.5a8) on all targeted genes across all samples. Pindel output was parsed using a custom Java program. The CNVs kept for further analyses were the high confidence calls: >8 reads supporting the CNV and the CNV can only be present in a maximum of 6 samples (some strains carry the same mutation). These

CNVs were read in via R, and compared the edited sequence, along with 20 bp of flanking region, to the ssODN sequence using pairwise alignment. In total, 1725 alleles were present in 278 samples. For 1409/1725 alleles, sequencing reads matched with ssODN sequences and were classified as 'disrupted' (Supplementary Data 1f). By allowing a maximum of 2 mismatches, another 99 editing events were identified and also classified as 'disrupted'. The remaining 217 alleles were manually curated through IGV (version 2.16.1)[70]. Small mutations, duplications, inversions, large deletions, or the combination of these events were detected (Supplementary Data 1f). These are classified as 'disrupted larger CNV' (Supplementary Data 1f). For two alleles yum1706 and yum2916 in which no mutations were detected, new genome editing was conducted and the disruption was confirmed using Sanger sequencing. The detailed sequence information can be found in Supplementary Data 1f.

## Behavioral analysis

To examine hypoxia-evoked locomotory response, OP50 was seeded on 5.5 cm assay plates. Bacteria were grown for 16 hours, and lawn border was removed before use. 25–30 day-one adults were transferred to assay plates and sealed in a microfluidic chamber. The defined gases were delivered into the microfluidic device with a flow rate of 3 ml/min by a syringe pump (PHD2000, Harvard Apparatus). The rapid switch between 7% and 1% $O_2$ was operated using Telfon valves (AutoMate Scientific) under the control of ValveBank Perfusion Controller. Videos were acquired with a FLIR camera mounted on a Zeiss Stemi-508 scope, and analyzed with a home-made MatLab program Zentracker [https://github.com/wormtracker/zentracker].

## Pathogen infection

A colony of bacteria (*V. cholerae* A1552, *P. aeruginosa* PA14 or *E. faecalis* OG1RF) was inoculated in 2 ml LB broth (A1552 and PA14) or BHI broth (OG1RF), and grown at 37 °C for 14 hours. The overnight culture was 1:100 diluted with either LB or BHI to 10 ml, and grown to OD = 2.0 at 37 °C. The pathogen was seeded to the center of NGM (A1552 and PA14) or BHI plates (OG1RF) to generate a pathogen lawn of 1.5 cm in diameter. These plates were incubated at 37 °C for 12 to 16 hours. 30 synchronized L4 animals fed with OP50 were washed twice in M9 and transferred to the pathogen plates. The assay was performed at room temperature. The number of animals remained on pathogen lawn was counted after 24 hours of pathogen exposure. The killing of *C. elegans* by pathogens was scored on daily basis, and animals were transferred to fresh pathogen plates every other day. Animals were deemed dead if they did not respond to prodding. In the screen, each mutant was assayed twice. If two data points were not consistent, a third assay was included. The candidates obtained were validated twice, each with three technical replicates. The avoidance index was calculated as (the number of animals out of the pathogen lawn / Total number of animals). The relative changes of pathogen avoidance were computed as (avoidance index of the mutant – avoidance index of wild type) / avoidance index of wild type. The occupancy was calculated as (the number of animals on the pathogen lawn / Total number of animals). The survival changes were calculated as (mean survival days of the mutant – mean survival days of wild type) / mean survival days of wild type.

To assay intestinal pathogen colonization, synchronized L4 animals were exposed to GFP-expressing *V. cholerae* for 16 hours, and then transferred to OP50 plates to clean the GFP-expressing bacteria from the skin for 5 minutes. GFP signals were visualized using Nikon A1 confocal microscope. To monitor the pharyngeal pumping, day-one adult animals of each strain were used for counting the number of contractions in the pharyngeal terminal bulb. One backward movement of the grinder was defined as one contraction. The pumping rates of 30 animals were recorded for each genotype.

## RNA-seq analysis

Twenty young adult animals were picked to fresh plates and allowed to lay eggs for 2 hours, after which adult animals were removed and eggs were allowed to develop into L4 stage. Ten plates of L4 animals were washed off and placed on either OP50 or *V. cholerae* plates for 8 hours. Subsequently, animals were collected and rinsed three times with M9 buffer. Worm pellets were frozen in liquid nitrogen. Animals were homogenized using Bullet Blender (Next Advance) in Qiazol Lysis Reagent with 0.5 mm Zirconia beads at 4 °C. RNA was prepared using RNeasy Plus Universal Mini Kit (Qiagen). Five independent RNA samples were prepared for both OP50 and *V. cholerae* treatment. Libraries were constructed and sequenced by Novogene. RNA-seq data were aligned using STAR v2.7.9a[71], and gene expression counts were extracted by featureCounts v2.0.3[72]. Differential expression was calculated using DESeq2[73]. GO analysis was performed using the EnrichR R package[74], with genes with *p* value < 1e-20. Gene expression counts for *P. aeruginosa*-infected *C.elegans* were obtained from a previous study[42].

## Chemotaxis assays

Chemotaxis assays for chemoreceptor mutants and in the screens were performed as outlined[53]. The 9 cm plates were filled with 10 ml of agar (1.7% agar, 1 mM $CaCl_2$, 1 mM $MgSO_4$, 25 mM potassium phosphate of pH 6). Two lines that were 1 cm apart were drawn on the plates, defining the area to place the animals (Fig. 4a). Two spots marked with plus on one side were designated for odorants, while two spots marked with dots on opposite side were designated for ethanol as control (Fig. 4a). Prior to the assay, 1 µl of 1 M NaN3 was added to all four spots. The attractants were prepared by diluting odorants with pure ethanol: diacetyl (1:2000), pyrazine (1:1000), TMT (1:2000), benzaldehyde (1:1000), 2-butanone (1:10000), IAA (1:200), and 2,3-pentanedione (1:10000). 1 µl of the attractant was added to each designated spot, and 1 µl of ethanol was placed on each spot on the other side. For undiluted repellents, different volumes of each odorant were used and equal volume of ethanol was used as control: diacetyl (5 µl), TMT (7.5 µl), benzaldehyde (2.5 µl), IAA (7.5 µl), 2,3-pentanedione (10 µl), 2-nonanone (1 µl) and 1-octanol (2.5 µl). 150–200 synchronized day-one adults were washed 3 times with chemotaxis buffer (1 mM $CaCl_2$, 1 mM $MgSO_4$, and 25 mM $K_2HPO_4$ pH 6), and placed to the center of assay plates, on which the odorants and ethanol were just spotted. The assays lasted for 1 hour and the plates were stored at 4 °C before counting. The chemotaxis indexes were calculated as (the number of animals of the odorant side – the number of animals on the ethanol side) / the total number of animals on both sides.

For *flp-19*, *flp-20* and *frpr-9* mutants, the chemotaxis assays were conducted as described[56]. The odorants and the ethanol were spotted 6 cm apart, and the animals were placed 4.5 cm away from odorant and ethanol spots (Supplementary Fig. 4a). The number of animals in 1 cm radius circles designated for odorant and ethanol were counted after 1 hour of chemotaxis. The rest of the procedures were identical to the aforementioned assays. The chemotaxis indexes were calculated as (the number of animals in the odorant circle – the number of animals in the ethanol circle) / the total number of animals on the plates.

For single worm assay[59], 10 ml medium was poured into the square assay plates. The plates were divided into six equal sectors (A to F), which were assigned with values 3, 2, 1, −1, −2, −3, respectively. Prior to the assays, the plates were air-dried for 1 hour. 1 µl of 2,3-pentanedione (1:10,000) was added onto each marked spot in A, and 1 µl of ethanol was added to each spot in F (Supplementary Fig. 6g). Animals were washed three time in chemotaxis buffer, and a single animal was placed to the plate center and assayed for 1 hour. Upon assay completion, the track of each animal on the assay plate was analyzed. Scores were assigned to the animal for each sector it visited, and the sum of scores were calculated and plotted for each animal.

## Microscopy

Worms were paralyzed with 50 mM NaN₃, and mounted on glass slides coated with 2% agarose in M9 buffer. The fluorescent images were acquired using Nikon A1 confocal microscope with Nikon NIS elements software. Synchronized day-one adults were used. Images were taken within 5 minutes after mounting. The images were analyzed using Fiji ImageJ.

## Calcium imaging

Calcium imaging was performed using modified custom microfluidic devices[75–77]. Images were collected using an Olympus BX52WI microscope with a 40x oil objective and Hamamatsu Orca CCD camera at 4 Hz with 4×4 binning. Odorants were diluted in filtered S-Basal buffer, and 1 µl of 20 µM fluorescein was added to visualize fluid flow. Worms were paralyzed in 10 mM (−)-tetramisole hydrochloride (levamisole) (Sigma L9756) prior to imaging. Odor-evoked calcium transients in AWA and AWC were recorded for 1 cycle of 30-second buffer/30-second odor/30-second buffer stimulus. Data were collected from biologically independent experiments over 2-3 days.

Following image acquisition, image slices were aligned using the Template Matching plugin in Fiji/ImageJ. Cell body and background regions of interests (ROIs) were defined manually, and background-subtracted fluorescence intensity values were used for analysis. To correct for photobleaching within each recording, an exponential decay was fit to fluorescence intensity values before and after stimulus presentation (first 30 seconds and the last 20 seconds per recording). The average $\Delta F/F_0$ value for 5 seconds before odor onset was calculated and set as $F_0$. All analysis, including the mean and standard error, were calculated and displayed using RStudio.

## Olfactory receptor expression in HEK293T cells

HEK293T cells (ATCC, CRL-11268) were cultured in Dulbecco's Modified Eagle's Medium (DMEM, Gibco) supplemented with 10% fetal bovine serum (Gibco) and 1% (v/v) penicillin-streptomycin (Gibco). Cells were plated at a density of $2.5 \times 10^5$/ml in 35 mm dishes for 24 hours. pcDNA3-HA-SRX-64 and pcDNA3-HA-SRX-2 constructs were transfected into HEK293T cells using polyethylenimine 25,000 (PEI, Polysciences, 1 mg/ml). After 16 hours of transfection, cells were split into 4-well plates (thermos fisher, 176740) with 50 µg/mL poly-D-lysine-coated glass coverslips at a density of $10^5$/ml, and incubated for another 8 hours for the attachment to coverslips. Cells were washed with prewarmed PBS and surface-stained with HA antibody (AS12 2220, Agrisera, 1:500 dilution) for 30 min at 37 °C. Receptor expression was measured by staining with the secondary antibody conjugated to Alexa 568 (A10042, Invitrogen, 1:1000 dilution in PBS) for 1 hour at room temperature. Cell nuclei were stained with 0.5 µM DAPI (diluted in PBS) for 15 min. Cells were imaged by Leica SP8 confocal microscope with a 63x oil objective.

## cAMP assay

HEK293T cells were plated at a density of $2.5 \times 10^5$/ml in 35-mm dishes 24 hours before transfection. pcDNA3 vector, pcDNA3-HA-SRX-64 and pcDNA3-HA-SRX-2 constructs were transfected into HEK293T cells using Lipofectamine™ 2000 (thermos fisher). The transiently transfected cells were cultured for 40 hours, subsequently detached from 35 mm dishes and resuspended in complete culture medium. Cells were then split into 96-well plates and incubated for 4–6 hours at 37 °C. The odorants were prepared with induction buffer (serum-free medium containing 500 µM IBMX and 100 µM Ro 20-1724). Cells were treated with 20 µl of chemical solutions at room temperature to initiate induction. cAMP levels were measured 10-15 minutes after ligand addition according to the protocol of cAMP-Glo™ Assay kit (Promega) and measured via Synergy 2 luminescence detection system (BioTek, Winooski, VT, USA).

## Statistical analysis and reproducibility

Sample sizes for all analyses are determined following established protocols and standard practice in *C. elegans*. No data were excluded. To ensure reproducibility, different alleles of each gene were used for phenotypic analyses, with data typically generated from three biological replicates, each containing at least three technical replicates. Statistical analyses were performed using GraphPad Prism 9, and the exact *p* values were listed in Source Data file. Data were presented as mean values +/- SEM with individual data points displayed. Blinding was applied to all the assays whenever possible.

## Reporting summary

Further information on research design is available in the Nature Portfolio Reporting Summary linked to this article.

## Data availability

The WGS sequencing data is available on ArrayExpress #E-MTAB-12983, and the RNA-seq data at ArrayExpress #E-MTAB-13001. Source data are provided with this paper.

## Code availability

Hypoxia-evoked locomotory response was analyzed using a custom-made MatLab program Zentracker [https://github.com/wormtracker/zentracker].

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

## Acknowledgements

We thank the Caenorhabditis Genetics Center (funded by NIH Office of Research Infrastructure Programs P40 OD010440) and the National BioResource Project Japan for strains. The computations were enabled by resources in project snic2022-5-18 provided by the Swedish National Infrastructure for Computing (SNIC) at UPPMAX, partially funded by the Swedish Research Council through grant agreement no. 2018-05973. This work was supported by the Swedish Research Council project grant (2018-02914) to S.N.W, the Medical Research Council (MC-A658-5TY30) and the HFSP Research Grant (RGP0001/2019) to A.B, the NIH (R35 GM122463) and the NSF (IOS 20421000) to P.S, the Swedish VR Research Council grant to MIMS (2021-06602), the ERC starting grant (802653 OXYGEN SENSING), the Swedish Research Council VR starting grant (2018-02216), and the Wallenberg Centre for Molecular Medicine (Umeå) to C.C.

## Author contributions

L.P., J.G., R.V.S., S.N.W., A.B., M.T., P.S., J.H. and C.C. designed experiments. L.P., J.W., Q.L., L.N., A.Ph., A.Pa., L.Z., R.V.S., A.K., T.V.P., W.H.H., S.L.M. and C.W. performed experiments, L.P., J.W., A.Ph., A.Pa., L.Z., R.V.S., A.B., M.T. P.S., J.H. and C.C. analyzed the data. R.V.S., J.G., S.N.W., A.B., P.S., J.H. and C.C. wrote and edited the paper.

## Funding

## Competing interests

The authors declare no competing interests.
