## [Peer Review File · Nature Communications]

Dissecting the genetic landscape of GPCR signaling through phenotypic profiling in *C. elegans*REVIEWER COMMENTS

Reviewer #1 (Remarks to the Author):

In this manuscript, Pu et al. develop a new approach and a set of resources to allow researchers to identify G protein coupled receptors (GPCRs) and neuropeptides that mediate almost any biological function in *C. elegans*. *C. elegans* has been an important model system for studying GPCR and neuropeptide signaling. However, progress has been stymied by the fact that there are >1,000 GPCR genes and >100 neuropeptide genes. Because there are so many of these genes and they often act redundantly, it has been excruciatingly difficult to assign GPCRs or neuropeptides to specific functions, or to match neuropeptides with their specific GPCR receptors. Pu et al. describe an exciting approach in which they used CRISPR-based gene editing to disrupt several closely-related GPCR or neuropeptide genes at a time in a single *C. elegans* strain. They did this on a genome-wide scale, generating 284 such strains that together contain disruptions of almost all GPCR and neuropeptide genes. They then screened these strains for defects in three different biological processes, and in each case identified specific GPCRs and/or neuropeptides that mediate those processes. Figures 2, 6, and 7 show particularly impressive results. The authors show that their approach can identify instances in which up to three GPCRs function redundantly, the type of thing that has been almost impossible to discover previously. They also show that their approach can match up putative neuropeptide/receptor pairs that function together. Finally, they show that their approach is successful in identifying specific chemoreceptors for volatile odorants, something that has been excruciatingly difficult or impossible in the past, despite much effort. The experimental work that supports the results described above is overall very impressive – the work is well done, and typically a well-controlled and complementary set of experiments are used to support each point made.

The writing in the manuscript is less impressive than are the experiments and results. I feel that the abstract and introduction, in particular, could be very much improved to better communicate the rationale for the work presented and the significance of the results. Below I detail some more specific criticisms of the presentation.

1. The abstract and introduction could have been much better written. It is difficult for the reader to understand what the issue is that needs to be resolved, exactly what the authors did to resolve it, and why this is so significant. The first paragraph of the Discussion, in contrast, provides an excellent summary of why this study is important. I'd love to see the authors redo their abstract and summary using some of the material from that Discussion paragraph.

An example of not very good writing in the abstract is “our resource allows us to identify sensory receptors, neuropeptides, and neuropeptide receptors in a single screen”. This is weird – they aren't actually identifying these molecules themselves, but rather identifying which known molecules that provide specific functions. Another example: the “two libraries” mentioned in the final sentence of the

abstract are the GPCR and neuropeptide mutant libraries, but that isn't explained clearly. Disrupting a lot of genes in a "limited number of strains" isn't a very clear idea – what does that mean and why is it important? I understood how brilliant all of this was by the time I finished reading the paper, but I absolutely did NOT understand the value of the paper after reading the abstract.

Specific comments:

1. Lines 99-101: "We extracted GPCR information from the *C. elegans* genome WS273 and later versions, and obtained a list of 1442 putative chemoreceptor encoding genes, among which 126 have not been annotated (Supplementary Table 1a)6-9". I do not understand how the authors "extracted GPCR information" from the *C. elegans* genome sequence. They do not describe this in their methods. The analysis of the 126 new GPCRs is central to Figure 1, and the authors need to describe their methods and criteria for identifying these previously unannotated GPCRs. Regarding the 126 previously unannotated chemoreceptor homologs the authors have identified encoded in the worm genome, they point us to Supplementary Table 1A for details. This table lists all 1442 putative chemoreceptors and apparently does not indicate which 126 of them are the novel ones that are the focus of Figure 1. The authors should indicate Supplementary Table 1A which putative chemoreceptors are novel. In line 104 the authors refer to "a couple of new members in the srw family". I infer they are referring to the six receptor homologs named in blue font in Figure 1C, but this is hard to figure out since I had to realize that by "a couple" the authors meant six and the Figure 1C legend provided almost no explanation of what I am looking at in the figure.

2. Lines 116-117: "We further identified 11 putative GPCRs that were annotated as belonging to the Chromadorea class of nematodes (Supplementary Table 1a)." I have no idea what the authors are trying to say with this sentence. These molecules are apparently GPCRs encoded by the *C. elegans* genome, and so how can these molecules "belong" to a class of different nematode species?

3. The paragraph starting on line 129 indicates in its bold-faced title as well as within the paragraph that the authors have generated "deletion" libraries of GPCR and neuropeptide genes. Indeed, the term "deletion library" is used throughout the manuscript. The authors should not use the term "deletion" to refer to their mutations. They can use a term like "gene disruption". In the *C. elegans* genetics field, when we talk about a "deletion" mutation we are talking about a mutation in which a large segment of the genomic DNA for a gene, often several hundred base pairs to several kilobases, containing multiple exons of the gene, have literally been deleted from the genome. Such deletion mutations have already been generated for most *C. elegans* genes, including a large set of the GPCR and neuropeptide genes studied in this manuscript. The new CRISPR-induced GPCR and neuropeptide gene mutations the authors have generated are not "deletion" mutations in this sense. The CRISPR technique the authors used induces a double stranded break in a targeted gene and uses a short oligonucleotide as a repair template to introduce a stop codon, a frame shift, and restriction site into one exon of the targeted gene. The distinction between a deletion mutation and the CRISPR-induced gene disruptions the authors induced is important. Since most *C. elegans* genes, including GPCR genes, are alternatively spliced, disrupting one

exon may only disrupt a subset of transcripts from a gene and thus may not be a null mutation. A large deletion of multiple exons is more likely to be a null mutation.

4. The paragraph starting on line 129 is confusing. It is hard to deduce from what is written that what the authors did was to generate 284 *C. elegans* strains, such that within each strain they have targeted one to 23 GPCR genes for disruption. Often they targeted a set of similar GPCRs within a single strain, but often the group of GPCRs targeted within a strain is more arbitrary. Overall, 1654 GPCR genes were targeted. I would like to see the writing in this paragraph clarified.

5. In the paragraph starting on line 152, and in Figure 1F, the authors provide a complicated discussion of whether the 1654 genes targeted for disruption were successfully disrupted or not. The conclusion after analysis of whole genome sequencing is that they are confident that “the majority” if not all of the genes have been disrupted. In fact, the authors show in Fig 1f that 131 genes had inconclusive results from genome sequencing, a further 11 appeared to be unedited, and then in the paragraph starting on line 152 they say that 26 of genes from these categories were subsequently disrupted. It seems like the authors have data that point specifically to 115 of the GPCR genes perhaps having NOT been disrupted successfully. The paper does not indicate in the supplemental tables or anywhere else which 115 genes these are. The data supplied in the paper should include this information.

6. In Supplementary Fig. 1, the authors should explain what the red segments on the horizontal axis are. I believe these are the time periods compared for statistical analysis, but the figure legend should tell me this rather than my having to guess it.

7. The text of the manuscript never refers to Fig. 1a. Typically, journals require that each figure is actually referred to in the text.

8. The results in Fig. 2 are really impressive and demonstrate the novel power of the approach used here to get around the redundancy of GPCRs and to help match GPCRs with neuropeptide ligands. Great! This is a significant and novel advance.

9. For the results in Figure 1, the authors start with a quintuple or quadruple mutants that show a hypoxia phenotype, and then proceed to test various strains containing subsets of the original mutations. The methods section does not specify how these strains were derived. I assume this was done by standard genetic crosses, for example crossing the original multiple mutant strains to the wild type and using PCR genotyping in subsequent generations to obtain new strains in which subsets of the original mutations were homozygosed. The methods section should state what strategy was used for these types of genetic experiments.

10. Figure 3a, Supplemental Fig. 2a-2d and Supplemental Table 4a all present a re-analysis of data on *C. elegans* gene expression changes in response to pathogen exposure that were generated and previously published by another lab. This gene expression analysis never comes up again in this manuscript, and one needs to read deep into the methods section to realize that the presentation shown is based on previously published data from another lab, not on data generated by the authors. One experiment related to this data done by the authors is shown in Supplemental Fig 2e, but again this never comes up again. I felt that the analyses in Fig 3a, Supplemental Fig. 2a-2e and Supplemental Table 4a were not inappropriate to include in this manuscript.

11. Lines 246-248: “We confirmed 9 genes that were disrupted in 10 strains, including *npr-1*, *flp-21*, *fmi-1*, *pdfr-1*, *pdf-1*; *pdf-2*, *dmsr-7*, *dop-6*, and *f59b2.13* (Fig. 3c, d and Supplementary Fig. 2g–n).” This is a poorly constructed and confusing sentence. The authors meant to say that single or double mutants for 9 of the mutations that were derived from 10 of the strains in the screen showed defects in pathogen response.

12. Lines 280-281 The sentence “The phenotypes were confirmed using *fmi-1* single mutants (Fig. 3d, i)” refers to the wrong figures. It should instead refer to Fig. 3i and Supplemental Fig. 3L.

13. I felt that the authors undersold some of the results presented in Fig. 4. They found that certain neuropeptide receptor and neuropeptide mutants have defects in chemotaxis to AWC-sensed odorants. On line 309, they cite reference 63 as having already shown this, but in fact the findings shown in the current manuscript are for different peptides and receptors than in reference 63. The authors on line 314 state that reference 15 had already shown that FLP-19 is a ligand for FRPR-9. However, reference 15 showed only that FLP-19 peptides can activate FRPR-9 in a heterologous cell assay, while the data presented in the current manuscript appear to demonstrate that this ligand/receptor pairing is actually physiologically relevant in *C. elegans* and show that it is involved in volatile odorant responses.

14. Figure 6 and Supplemental Fig 6 showed a really impressive result demonstrating that 3 chemoreceptors redundantly mediate response to 2,3-pentanedione. These figures presented a really lovely set of complementary experiments that together nailed the findings. Given how difficult it has been to match odorants with receptors in *C. elegans*, this is truly impressive and demonstrates the potential of screening the library of GPCR mutants generated in this paper.

15. Lines 398-399 “We also found the strain CHS1146, which exhibited defects in chemotaxis to low concentrations of the volatile odorant pyrazine” should read instead “We also found that strain CHS1146 exhibited defects in chemotaxis to low concentrations of the volatile odorant pyrazine. (Fig. 4d)”

16. Lines 411-413 “Moreover, expressing *srx-64* under the *odr-10* promoter restored pyrazine chemotaxis to *srx-64* mutants (Fig. 7g). These observations suggest that SRX-64 acts in AWA neurons for the sensation of pyrazine.” Instead of making this a puzzle that is difficult for the reader to solve, the authors should have just told us that *odr-10* promoter is active exclusively in AWA neurons. There are other examples like this in the manuscript where the authors write as though they assume the reader is already an expert in fine details of their experimental system. Instead, the paper should be written for a more general audience.

Reviewer #2 (Remarks to the Author):

In their study, Pu et al. employed CRISPR/Cas9 genome editing to construct GPCR and neuropeptide deletion libraries in *C. elegans*. They disrupted a total of 1654 GPCR genes across 284 strains, as well as 152 neuropeptide encoding genes in 38 strains. They then conducted screenings on these mutant libraries to identify strains that exhibited impaired responses to acute hypoxia, pathogens, and volatile odorants.

Regarding acute hypoxia response, they examined all mutants in the deletion libraries and discovered that the neuropeptide FLP-1 potentially regulates locomotory responses to acute hypoxia via DMSR-4, 7, and 8. Moving on to pathogen defense mechanisms, they screened both GPCR and neuropeptide mutants and identified new regulators, such as FRPR-9, involved in behavioral avoidance and innate immune responses to bacterial pathogens.

Furthermore, the researchers assessed the mutant strains from the two libraries for their responses to odorants. Through this screening, they successfully identified SRX-2 and SRX-64 as receptors for 2,3-pentanedione and pyrazine, respectively. They further confirmed the role of SRX-2 and SRX-64 in chemosensation via calcium imaging, behavioral assays and heterologous expression.

The creation of these two mutant libraries presents a highly valuable platform for future investigations into the neural mechanisms underlying information encoding, signal transduction, and behavioral processes. As a result, this manuscript is overall interesting, demonstrating meticulous methodology and producing reliable experimental outcomes.

The basic conclusions of this study are well supported by the physiological, behavioral, and genetic experiments. However, there are certain aspects of the manuscript that would benefit from further clarification.

Previous studies have already reported the interaction between FLP-1 and DMSR-7, which somewhat diminishes the novelty of identifying FLP-1's interaction with DMSR-4, 7, and 8 in response to acute hypoxia. Additionally, considering that FLP-1 and DMSR-4, 7, 8 regulate the acute response to hypoxia, an explanation is needed as to why they also play a role in regulating basal locomotion under 7% O₂ conditions.

While the authors successfully identified several new regulators involved in behavioral avoidance and innate immune responses to gram-negative bacterial pathogens, such as FRPR-9, there is a lack of explanation or discussion regarding how these genes specifically contribute to the response to *V. cholerae*. Further clarification and elaboration are needed.

While it is plausible that SRX-2 and SRX-64 could serve as receptors for 2,3-pentanedione and pyrazine, respectively, the methods used by the authors to validate this claim are somewhat confusing. The authors performed ectopic expression of SRX-2 and SRX-64 in HEK293 cells, followed by the assessment of intracellular cAMP levels upon the application of odorants. However, further clarification, elaboration, and references are needed to support and explain this experimental approach in more detail.

The n values are not clearly indicated in the manuscript. Additionally, figures 6e, 6k, 7e, 7j, S5n, and S6c lack a scale bar on the micrographs.

Reviewer #3 (Remarks to the Author):

In the manuscript by Pu et al., the authors employed CRISPR technology to disrupt a majority of GPCR and neuropeptide encoding genes in *C. elegans*. This approach allowed them to generate multiple genes deletions within a single strain, enabling the identification of genes with redundant functions. Additionally, the authors conducted three screens to identify neuropeptides and receptors that play essential roles in hypoxia-evoked locomotory responses, pathogen-induced immune response, and detection of volatile food-related odorants. Overall, the experiments were meticulously executed and well-organized. These three screens help to a better understanding of the GPCR signaling in multiple processes, and provide valuable resources for the worm community. However, before publication, the following concerns need to be addressed.

Major:

1. The manuscript lacks a detailed description of the methodology employed to achieve multiple gene disruptions in each strain using the CRISPR/Cas9 system. This information is crucial, especially considering the emphasis on identifying genes with redundant functions. Additionally, the authors did

not provide a specific explanation of the strategy used to generate and validate the homologous deletions of multiple genes. These aspects need to be addressed in order to enhance the clarity and completeness of the manuscript.

2. In Supplementary Figure 1a, it is observed that the quintuple mutant, with disruptions in five genes (*dmsr-4*, *dmsr-5*, *dmsr-6*, *dmsr-7*, and *dmsr-8*), exhibited defects in the O₂-regulated locomotor behavior. However, the authors did not provide an explanation for their focus on the three mutations (*dmsr-4*, *dmsr-7*, and *dmsr-8*) or whether *dmsr-5* and *dmsr-6* are also involved in the hypoxia-evoked locomotory response. Clarification regarding these aspects is necessary to enhance the understanding of the study findings.

3. In figure 5c, whether there are significant differences between the *flp-19;flp-20* mutant and the *frpr-9;flp-19;flp-20* mutant, as well as between the *frpr-19* mutant and the *frpr-9;flp-19;flp-20* mutant?

4. In Figure 5g and h, it is intriguing to observe that the expression of *flp-19* alone was able to rescue the defect in 2,3-pentanedione chemotaxis observed in *flp-19; flp-20* double mutants. The authors should thoroughly discuss the potential mechanisms underlying this phenomenon.

5.

In Figure 6g, the authors demonstrated that overexpressing *srx-1* and *srx-3* in the AWCOFF neuron successfully rescued the reduced chemotaxis observed in *srx-2* mutants, although *SRX-2* appeared to play a more prominent role. Interestingly, the authors also noted that the expression levels of *srx-1* and *srx-3* were significantly lower in AWCOFF compared to *srx-2*. This raises the question of whether the higher expression level of *srx-2* determines its major role. The authors should discuss the potential relevant functions of the three proteins in more detail. Additionally, it is important to address why the AWCOFF neuron requires three genes encoding 2,3-pentanedione receptors and the significance of this observation.

6. Whether ectopic expression in AWB together with *srx-1* and *srx-2* can increase the aversive responses to 2,3-pentanedione?

Minor:

1. In figure 2c, the authors stated that by examining the responses of each single mutant, we found that mutating *flp-1* yielded a phenotype similar to that of the quadruple mutants. It is better to include the data of other single mutants (*flp-14*, *flp-23*, and *flp-25*) in the supplementary figures.

2. In Supplementary Figure 2a and b, the different colors represent distinct categories of genes. However, in Supplementary Figure 2c and d, the same set of colors represents different levels of significance. This can potentially lead to confusion.

3. In Figure 4e, it would be beneficial to include information about the specific chemosensory neurons that are involved in detecting individual chemicals. This additional information will provide a more comprehensive understanding of the neuronal basis for the observed responses.

4. In line 318, "AWC mediated" should be "AWC-mediated".

REVIEWER COMMENTS

Reviewer #1 (Remarks to the Author):

In this manuscript, Pu et al. develop a new approach and a set of resources to allow researchers to identify G protein coupled receptors (GPCRs) and neuropeptides that mediate almost any biological function in *C. elegans*. *C. elegans* has been an important model system for studying GPCR and neuropeptide signaling. However, progress has been stymied by the fact that there are >1,000 GPCR genes and >100 neuropeptide genes. Because there are so many of these genes and they often act redundantly, it has been excruciatingly difficult to assign GPCRs or neuropeptides to specific functions, or to match neuropeptides with their specific GPCR receptors. Pu et al. describe an exciting approach in which they used CRISPR-based gene editing to disrupt several closely-related GPCR or neuropeptide genes at a time in a single *C. elegans* strain. They did this on a genome-wide scale, generating 284 such strains that together contain disruptions of almost all GPCR and neuropeptide genes. They then screened these strains for defects in three different biological processes, and in each case identified specific GPCRs and/or neuropeptides that mediate those processes. Figures 2, 6, and 7 show particularly impressive results. The authors show that their approach can identify instances in which up to three GPCRs function redundantly, the type of thing that has been almost impossible to discover previously. They also show that their approach can match up putative neuropeptide/receptor pairs that function together. Finally, they show that their approach is successful in identifying specific chemoreceptors for volatile odorants, something that has been excruciatingly difficult or impossible in the past, despite much effort. The experimental work that supports the results described above is overall very impressive – the work is well done, and typically a well-controlled and complementary set of experiments are used to support each point made.

The writing in the manuscript is less impressive than are the experiments and results. I feel that the abstract and introduction, in particular, could be very much improved to better communicate the rationale for the work presented and the significance of the results. Below I detail some more specific criticisms of the presentation.

We thank our reviewer for the encouraging and supportive comments, and for the valuable suggestions to improve this manuscript.

1. The abstract and introduction could have been much better written. It is difficult for the reader to understand what the issue is that needs to be resolved, exactly what the authors did to resolve it, and why this is so significant. The first paragraph of the Discussion, in contrast, provides an excellent summary of why this study is important. I'd love to see the authors redo their abstract and summary using some of the material from that Discussion paragraph. An example of not very good writing in the abstract is "our resource allows us to identify sensory receptors, neuropeptides, and neuropeptide receptors in a single screen". This is weird – they aren't actually identifying these molecules themselves, but rather identifying which known molecules that provide specific functions. Another example: the "two libraries" mentioned in the final sentence of the abstract are the GPCR and neuropeptide mutant libraries, but that isn't explained clearly. Disrupting a lot of genes in a "limited number of strains" isn't a very clear idea – what does that mean and why is it important? I understood how brilliant all of this was by the time I finished reading the paper, but I absolutely did NOT understand the value of the paper after reading the abstract.

Many thanks for the suggestions, and our apology for the poorly constructed sentences, abstract and introduction. We have now re-written the whole abstract (Lines 29–49) and parts of

the introduction (Lines 81–91). As suggested, we used some of the contents from the first paragraph of the discussion in the new abstract and introduction, trying to highlight the significance of this study.

Specific comments:

1. Lines 99-101: “We extracted GPCR information from the *C. elegans* genome WS273 and later versions, and obtained a list of 1442 putative chemoreceptor encoding genes, among which 126 have not been annotated (Supplementary Table 1a)6-9“. I do not understand how the authors “extracted GPCR information” from the *C. elegans* genome sequence. They do not describe this in their methods. The analysis of the 126 new GPCRs is central to Figure 1, and the authors need to describe their methods and criteria for identifying these previously unannotated GPCRs. Regarding the 126 previously unannotated chemoreceptor homologs the authors have identified encoded in the worm genome, they point us to Supplementary Table 1A for details. This table lists all 1442 putative chemoreceptors and apparently does not indicate which 126 of them are the novel ones that are the focus of Figure 1. The authors should indicate Supplementary Table 1A which putative chemoreceptors are novel. In line 104 the authors refer to “a couple of new members in the *srw* family”. I infer they are referring to the six receptor homologs named in blue font in Figure 1C, but this is hard to figure out since I had to realize that by “a couple” the authors meant six and the Figure 1C legend provided almost no explanation of what I am looking at in the figure.

We thank our reviewer for the suggestion. We have now thoroughly described how we obtained the list of GPCR encoding genes in the methods section (Lines 514–524). Briefly, we obtained the full list of annotated GPCR encoding genes in each subfamily by searching them under the directory of ‘gene class’ using GPCR subfamily names (e.g., *sra*) on Wormbase homepage. Second, the paralogs of each GPCR encoding gene were obtained from the gene information page in Wormbase. Third, the protein sequences of all the GPCRs were used as the templates to BLAST search for similar proteins in the Wormbase (Lines 514–524 for details).

We apologize for the poor presentation of the 126 new GPCRs, which are now listed in a new excel sheet (Supplementary Table 1b). For the six *srw* genes, we explicitly state that ‘Six unannotated members of the *srw* family are closely related to neuropeptide receptors (Fig. 1c)’ (Lines 109–110), and highlighted these six genes in Supplementary Table 1b. We also revised the figure legend accordingly (Lines 1052–1058).

2. Lines 116-117: “We further identified 11 putative GPCRs that were annotated as belonging to the Chromadorea class of nematodes (Supplementary Table 1a).” I have no idea what the authors are trying to say with this sentence. These molecules are apparently GPCRs encoded by the *C. elegans* genome, and so how can these molecules “belong” to a class of different nematode species?

Our apology for this confusing sentence. It has been rephrased as the following ‘Additionally, we identified 11 nematode-specific GPCRs, which shared significant sequence similarity and were clustered closely with chemoreceptors’ (Lines 117–119). However, these 11 receptors were classified as ‘GPCRs Chromadorea’ in the Wormbase. We still indicate them with this name in fig 1c and in the Supplementary Table 1a.

3. The paragraph starting on line 129 indicates in its bold-faced title as well as within the paragraph that the authors have generated “deletion” libraries of GPCR and neuropeptide genes. Indeed, the term “deletion library” is used throughout the manuscript. The authors should

not use the term “deletion” to refer to their mutations. They can use a term like “gene disruption”. In the *C. elegans* genetics field, when we talk about a “deletion” mutation we are talking about a mutation in which a large segment of the genomic DNA for a gene, often several hundred base pairs to several kilobases, containing multiple exons of the gene, have literally been deleted from the genome. Such deletion mutations have already been generated for most *C. elegans* genes, including a large set of the GPCR and neuropeptide genes studied in this manuscript. The new CRISPR-induced GPCR and neuropeptide gene mutations the authors have generated are not “deletion” mutations in this sense. The CRISPR technique the authors used induces a double stranded break in a targeted gene a uses a short oligonucleotide as a repair template to introduce a stop codon, a frame shift, and restriction site into one exon of the targeted gene. The distinction between a deletion mutation and the CRISPR-induced gene disruptions the authors induced is important. Since most *C. elegans* genes, including GPCR genes, are alternatively spliced, disrupting one exon may only disrupt a subset of transcripts from a gene and thus may not be a null mutation. A large deletion of multiple exons is more likely to be a null mutation.

We thank our reviewer to clarify this. Our resources are now referred as ‘GPCR and neuropeptide mutant collections or mutant libraries’, and the word ‘deletion’ has been replaced with ‘mutation’ or ‘disruption’ throughout the text.

4. The paragraph starting on line 129 is confusing. It is hard to deduce from what is written that what the authors did was to generate 284 *C. elegans* strains, such that within each strain they have targeted one to 23 GPCR genes for disruption. Often they targeted a set of similar GPCRs within a single strain, but often the group of GPCRs targeted within a strain is more arbitrary. Overall, 1654 GPCR genes were targeted. I would like to see the writing in this paragraph clarified.

We agree with our reviewer that the description of this part in our initial submission was insufficient. To address this concern, we added a new section to the methods, describing how the GPCR encoding genes were clustered into 284 subgroups for disruption (Lines 526–545). This description was not included in the main text due to space limitations. As pointed out by our reviewer, the clustering of a large proportion of GPCR-encoding genes was arbitrary. We acknowledge that our clustering approach is far from optimal, and the strains we generated can be further optimized. This caveat is briefly mentioned in the discussion (Lines 462–464).

The main criterion we used to cluster GPCRs is their protein sequence identity of >40%, which generated groups of varying sizes, ranging from 1 to 21 genes. In addition, their genetic locations and phylogenic relationships were considered. As noticed, the remaining GPCRs were arbitrarily assigned into different groups, while we took a few factors into consideration. We aimed to minimize the number of genes disrupted in each strain. First, it was uncertain if multiple rounds of CRISPR/Cas9 genome editing would significantly increase nonspecific mutations or cause genome rearrangement in the background. Second, multiple rounds of genome editing in the same strain were highly time-consuming. To ensure time efficiency, most strains were subject to a maximum of 2 rounds of micro-injections (≤ 6 genes) (Supplementary Table 1e). Third, GPCRs from different sub-families were typically not clustered in the same group and were not disrupted in the same strain. For example, *sro-1* was the only gene disrupted in the strain CHS1013 (Supplementary Table 1e). Moreover, genes with well-established functions, such as *odr-10*, were typically kept separate from others.

We hope our reviewer will find the new section in the methods to be useful (Lines 526–545).

5. In the paragraph starting on line 152, and in Figure 1F, the authors provide a complicated discussion of whether the 1654 genes targeted for disruption were successfully disrupted or not. The conclusion after analysis of whole genome sequencing is that they are confident that “the majority” if not all of the genes have been disrupted. In fact, the authors show in Fig 1f that 131 genes had inconclusive results from genome sequencing, a further 11 appeared to be unedited, and then in the paragraph starting on line 152 they say that 26 of genes from these categories were subsequently disrupted. It seems like the authors have data that point specifically to 115 of the GPCR genes perhaps having NOT been disrupted successfully. The paper does not indicate in the supplemental tables or anywhere else which 115 genes these are. The data supplied in the paper should include this information.

Regretfully, the presentation of genome sequencing analysis was inadequate in our initial submission. Most inconclusive disruptions were attributed to low sequencing depth. To address this concern, we requested our sequencing provider to re-sequence those libraries with poor coverage in the first round of genome sequencing. Following the completion of the second round of sequencing, we combined the sequencing data and did the analysis again. We confirmed that all targeted genes have been successfully edited. The sequencing information of each allele and its comparison to ssODN template were provided in Supplementary Table 1f. As a result, the complicated discussion was omitted (Lines 149–154). We think that the original fig 1e and 1f for genome sequencing analysis are unnecessary, and have been removed from fig 1.

6. In Supplementary Fig. 1, the authors should explain what the red segments on the horizontal axis are. I believe these are the time periods compared for statistical analysis, but the figure legend should tell me this rather than my having to guess it.

We apologize for this. It has now been fixed as suggested (Lines 1308–1309).

7. The text of the manuscript never refers to Fig. 1a. Typically, journals require that each figure is actually referred to in the text.

We thank our reviewer for discovering this. Fig. 1a has now been cited in the text (Lines 94–95, and line 159).

8. The results in Fig. 2 are really impressive and demonstrate the novel power of the approach used here to get around the redundancy of GPCRs and to help match GPCRs with neuropeptide ligands. Great! This is a significant and novel advance.

We thank our reviewer for the encouraging comment.

9. For the results in Figure 1, the authors start with a quintuple or quadruple mutants that show a hypoxia phenotype, and then proceed to test various strains containing subsets of the original mutations. The methods section does not specify how these strains were derived. I assume this was done by standard genetic crosses, for example crossing the original multiple mutant strains to the wild type and using PCR genotyping in subsequent generations to obtain new strains in which subsets of the original mutations were homozygosed. The methods section should state what strategy was used for these types of genetic experiments.

Our reviewer is correct. We typically identify the causal mutations by genetic crosses. But there are exceptions. When two or more disrupted genes within a strain are genetically linked, we

often generate new alleles using CRISPR/Cas9, either independently or in combination. This is now described as follows in the methods (Lines 483–490):

Strains in the mutant collections usually contain multiple mutations. To identify the causal mutations responsible for the phenotypes, the original strains were typically crossed with the wild type (N2), and the offspring in the subsequent generations were genotyped to obtain a set of strains with different combinations of mutations. These mutants were assessed for their responses to the relevant stimuli. If the mutations are genetically linked, new alleles of these genes were generated using CRISPR/Cas9, either independently or in combination. The resulting strains were subject to phenotypic analyses.

10. Figure 3a, Supplemental Fig. 2a-2d and Supplemental Table 4a all present a re-analysis of data on *C. elegans* gene expression changes in response to pathogen exposure that were generated and previously published by another lab. This gene expression analysis never comes up again in this manuscript, and one needs to read deep into the methods section to realize that the presentation shown is based on previously published data from another lab, not on data generated by the authors. One experiment related to this data done by the authors is shown in Supplemental Fig 2e, but again this never comes up again. I felt that the analyses in Fig 3a, Supplemental Fig. 2a-2e and Supplemental Table 4a were not inappropriate to include in this manuscript.

We apologize for the inadequate writing and lack of clarity throughout the text. In fact, we conducted RNA-sequencing on animals that were exposed to *V. Cholerae*. As noticed by our reviewer, we did not perform RNA-seq on animals that were exposed to PA14, since this has been explored by other laboratories on several occasions. In supplementary Fig. 2a and 2b, we conducted a comparative analysis of our RNA-seq data with the published PA14 dataset. The rest of figure panels including Fig. 3a and Supplementary Fig. 2c-2e was derived from our own sequencing data. We anticipate our reviewer will agree that it is appropriate to include these data in the manuscript. To avoid any confusion, we have included a more explicit description of the data in the text (Lines 205–207). The raw sequencing data have been uploaded and can be accessed on ArrayExpress #E-MTAB-12983.

11. Lines 246-248: “We confirmed 9 genes that were disrupted in 10 strains, including *npr-1*, *flp-21*, *fmi-1*, *pdfr-1*, *pdf-1*; *pdf-2*, *dmsr-7*, *dop-6*, and *f59b2.13* (Fig. 3c, d and Supplementary Fig. 2g–n).” This is a poorly constructed and confusing sentence. The authors meant to say that single or double mutants for 9 of the mutations that were derived from 10 of the strains in the screen showed defects in pathogen response.

Our apology for the poorly constructed sentence. We have now rephrased it.

‘In 10 strains defective in pathogen avoidance, we confirmed the functional importance of 9 genes including *npr-1*, *flp-21*, *fmi-1*, *pdfr-1*, *pdf-1*; *pdf-2*, *dmsr-7*, *dop-6*, and *F59B2.13*’ (Lines 226–228).

12. Lines 280-281 The sentence “The phenotypes were confirmed using *fmi-1* single mutants (Fig. 3d, i)” refers to the wrong figures. It should instead refer to Fig. 3i and Supplemental Fig. 3L.

We have fixed this (Line 261).

13. I felt that the authors undersold some of the results presented in Fig. 4. They found that

certain neuropeptide receptor and neuropeptide mutants have defects in chemotaxis to AWC-sensed odorants. On line 309, they cite reference 63 as having already shown this, but in fact the findings shown in the current manuscript are for different peptides and receptors than in reference 63. The authors on line 314 state that reference 15 had already shown that FLP-19 is a ligand for FRPR-9. However, reference 15 showed only that FLP-19 peptides can activate FRPR-9 in a heterologous cell assay, while the data presented in the current manuscript appear to demonstrate that this ligand/receptor pairing is actually physiologically relevant in *C. elegans* and show that it is involved in volatile odorant responses.

We thank our reviewer for this. As suggested, we revised the text to reiterate the significance of our observations.

For the sentence in line 309 (New lines 291–294), we wrote ‘Screens using the diluted odorants identified three peptidergic mutants (CHS1025, CHS10063 and CHS10013) that exhibited significantly reduced chemotaxis to all AWC– but not AWA– sensed odorants (Fig. 4d, e and Supplementary Table 5a), suggesting that peptidergic signaling modulates the AWC circuit (Chalasanani et al., 2010)’.

For the *frpr-9* and *flp-19* data (Lines 296–299), we revised the sentence as follows. ‘The strain CHS1025 disrupted the neuropeptide receptor encoding gene *frpr-9*, whereas CHS10063 contained a mutation in the neuropeptide encoding gene *flp-19* (Fig. 4e). These observations suggest that FLP-19 may act through FRPR-9 to regulate AWC-mediated chemosensation.’

14. Figure 6 and Supplemental Fig 6 showed a really impressive result demonstrating that 3 chemoreceptors redundantly mediate response to 2,3-pentanedione. These figures presented a really lovely set of complementary experiments that together nailed the findings. Given how difficult it has been to match odorants with receptors in *C. elegans*, this is truly impressive and demonstrates the potential of screening the library of GPCR mutants generated in this paper.

We thank our reviewer for the encouraging comments.

15. Lines 398-399 “We also found the strain CHS1146, which exhibited defects in chemotaxis to low concentrations of the volatile odorant pyrazine” should read instead “We also found that strain CHS1146 exhibited defects in chemotaxis to low concentrations of the volatile odorant pyrazine. (Fig. 4d)”

We have modified the sentence as suggested (Lines 402–403).

16. Lines 411-413 “Moreover, expressing *srx-64* under the *odr-10* promoter restored pyrazine chemotaxis to *srx-64* mutants (Fig. 7g). These observations suggest that SRX-64 acts in AWA neurons for the sensation of pyrazine.” Instead of making this a puzzle that is difficult for the reader to solve, the authors should have just told us that *odr-10* promoter is active exclusively in AWA neurons. There are other examples like this in the manuscript where the authors write as though they assume the reader is already an expert in fine details of their experimental system. Instead, the paper should be written for a more general audience.

We agree with our reviewer that there are multiple places in the manuscript where readability needs improvement for a general audience. The sentence related to the *odr-10* promoter has been updated as suggested (Lines 413–414). In addition, we have provided the description for a set of terms, such as GCaMP3, GCaMP2, and others.

Reviewer #2 (Remarks to the Author):

In their study, Pu et al. employed CRISPR/Cas9 genome editing to construct GPCR and neuropeptide deletion libraries in *C. elegans*. They disrupted a total of 1654 GPCR genes across 284 strains, as well as 152 neuropeptide encoding genes in 38 strains. They then conducted screenings on these mutant libraries to identify strains that exhibited impaired responses to acute hypoxia, pathogens, and volatile odorants.

Regarding acute hypoxia response, they examined all mutants in the deletion libraries and discovered that the neuropeptide FLP-1 potentially regulates locomotory responses to acute hypoxia via DMSR-4, 7, and 8. Moving on to pathogen defense mechanisms, they screened both GPCR and neuropeptide mutants and identified new regulators, such as FRPR-9, involved in behavioral avoidance and innate immune responses to bacterial pathogens.

Furthermore, the researchers assessed the mutant strains from the two libraries for their responses to odorants. Through this screening, they successfully identified SRX-2 and SRX-64 as receptors for 2,3-pentanedione and pyrazine, respectively. They further confirmed the role of SRX-2 and SRX-64 in chemosensation via calcium imaging, behavioral assays and heterologous expression.

The creation of these two mutant libraries presents a highly valuable platform for future investigations into the neural mechanisms underlying information encoding, signal transduction, and behavioral processes. As a result, this manuscript is overall interesting, demonstrating meticulous methodology and producing reliable experimental outcomes.

The basic conclusions of this study are well supported by the physiological, behavioral, and genetic experiments. However, there are certain aspects of the manuscript that would benefit from further clarification.

We thank our reviewer for the encouraging and supportive feedback, as well as for providing valuable suggestions to improve this manuscript.

Previous studies have already reported the interaction between FLP-1 and DMSR-7, which somewhat diminishes the novelty of identifying FLP-1's interaction with DMSR-4, 7, and 8 in response to acute hypoxia. Additionally, considering that FLP-1 and DMSR-4, 7, 8 regulate the acute response to hypoxia, an explanation is needed as to why they also play a role in regulating basal locomotion under 7% O₂ conditions.

We agree with our reviewer that the novelty of our discovery was compromised by the reported interaction between FLP-1 and DMSR-7. We think it is worth including this screen in the paper as a compelling example of identifying functionally redundant genes using our mutant collections.

Additionally, a potential rationale for the increased basal locomotion of *flp-1* mutants is provided in Lines 185–187.

While the authors successfully identified several new regulators involved in behavioral avoidance and innate immune responses to gram-negative bacterial pathogens, such as FRPR-

9, there is a lack of explanation or discussion regarding how these genes specifically contribute to the response to *V. cholerae*. Further clarification and elaboration are needed.

As suggested, we now provide several plausible explanations of how the novel regulators contribute to the response to *V. cholerae*.

1. NLP-40/AEX-2: They have been shown to play a role in the regulation of defecation cycle, implying that inefficient removal of pathogens from the intestine may underlie their increased sensitivity to the pathogen (Lines 247–250).
2. C11H1.2 and Y75B8A.16: They act to acidify Golgi luminal pH. The double mutants likely have defects in Golgi luminal acidification, which may damage the transport of proteins crucial for the defense against the infection (Lines 252–254).
3. FRPR-9/FLP-19: we performed additional experiments, excluding the possibility that the resistance of *frpr-9* mutants to *V. cholerae* is caused by reduced pathogen uptake or diminished intestinal pathogen accumulation (Supplementary Fig. 3s–v). We also found that *frpr-9* mutants were not only resistant to gram-negative bacterial pathogens *V. cholerae* and *P. aeruginosa*, but also exhibited enhanced tolerance to the gram-positive bacterium *E. faecalis* (Supplementary Fig. 3w, x), implying that FRPR-9-mediated peptidergic signaling likely modulates innate immunity in *C. elegans* (Lines 270–276).
4. DOP-6, F59B2.13 and FMI-1: FMI-1 has a well-established role in neuronal development, and may be required for the neuronal responses to bacterial infection (Lines 235–238 and Lines 261–264). Since *daf-7* mutants are defective in pathogen avoidance and *daf-7* expression is robustly induced by *P. aeruginosa* in a G-protein dependent manner (Meisel et al., 2014), we are currently investigating if the GPCR mutants isolated from our screen including *fmi-1*, *dop-6*, and *F59B2.13*, which are defective in pathogen avoidance, have defects in the activation of *daf-7* expression by the pathogen.

While it is plausible that SRX-2 and SRX-64 could serve as receptors for 2,3-pentanedione and pyrazine, respectively, the methods used by the authors to validate this claim are somewhat confusing. The authors performed ectopic expression of SRX-2 and SRX-64 in HEK293 cells, followed by the assessment of intracellular cAMP levels upon the application of odorants. However, further clarification, elaboration, and references are needed to support and explain this experimental approach in more detail.

Our apology for the confusion caused by the text. We have now elaborated on the rationale for using the cAMP assay to verify the interactions between odorants and receptors in HEK293 cells, and cited the relevant references (Lines 388–392). cAMP assays have been widely used to monitor the activation of odorant receptors expressed in heterologous cell lines including HEK293 and HEK293T-derived Hana3A cells. The odorant binding to the olfactory receptor induces the conformation changes of the receptor, which in turn binds and activates Gs, leading to an increased production of intracellular cAMP. This assay has also been used to assess a putative odorant receptor from *C. elegans* (Choi et al., 2022).

The n values are not clearly indicated in the manuscript. Additionally, figures 6e, 6k, 7e, 7j, S5n, and S6c lack a scale bar on the micrographs.

We thank our reviewer for pointing this out. We have now provided the n values to the relevant figure plots. The number of biological and technical replicates for the pathogen and chemotaxis

assays were described in the figure legends. The number of animals used for each strain in each assay can be found in the source data.

As suggested, we have inserted scale bars to the micrographs in figs. 6e, 6k, 7e, 7j, S5n, and S6c.

Reviewer #3 (Remarks to the Author):

In the manuscript by Pu et al., the authors employed CRISPR technology to disrupt a majority of GPCR and neuropeptide encoding genes in *C. elegans*. This approach allowed them to generate multiple genes deletions within a single strain, enabling the identification of genes with redundant functions. Additionally, the authors conducted three screens to identify neuropeptides and receptors that play essential roles in hypoxia-evoked locomotory responses, pathogen-induced immune response, and detection of volatile food-related odorants. Overall, the experiments were meticulously executed and well-organized. These three screens help to a better understanding of the GPCR signaling in multiple processes, and provide valuable resources for the worm community. However, before publication, the following concerns need to be addressed.

We thank our reviewer for the encouraging and supportive comments, and for the valuable suggestions to improve this manuscript.

Major:

1. The manuscript lacks a detailed description of the methodology employed to achieve multiple gene disruptions in each strain using the CRISPR/Cas9 system. This information is crucial, especially considering the emphasis on identifying genes with redundant functions. Additionally, the authors did not provide a specific explanation of the strategy used to generate and validate the homologous deletions of multiple genes. These aspects need to be addressed in order to enhance the clarity and completeness of the manuscript.

We apologize for omitting this in our initial submission. To address this, we first provided a detailed description of how to cluster different GPCR genes into subgroups, which can be found in Lines 526–545. However, the clustering of a large proportion of GPCR-encoding genes was arbitrary. We also acknowledge that our clustering approach is far from optimal, and the strains we generated can be further optimized. This caveat is briefly mentioned in the discussion (Lines 462–464).

Once a set of genes were assigned into a distinct group, we conducted multiple rounds of genome editing to disrupt all these genes in a single strain (Lines 547–589). Briefly, we targeted at three genes in the first round of injection, and 24 transgenic F1 animals were singled. To genotype the possible editing at each targeted site, two primers for each gene were used to amplify a fragment of 400 to 1000bp. The PCR products were digested with the relevant restriction enzymes. F1 rollers that were heterozygous mutants for all three targeted sites were retained to obtain homozygous mutants in their subsequent generations. Multiple rounds of singling and genotyping might be required. After obtaining the triple homozygous mutants, we proceeded with the next round of micro-injection to disrupt three additional genes in the mutant background. The same procedure was used to genotype the disruption of three new genes. This process was repeated until all the genes within the group were disrupted. Upon completion, all the editing events in each strain were validated again using PCR-based genotyping, and the

final strains were genome-sequenced. The detailed sequencing information at the targeted sites was displayed in Supplementary Table 1f.

We have expanded the section of CRISPR/Cas9 genome editing and explicitly described the methodology employed to achieve multiple gene disruptions in each strain (Lines 547–589).

2. In Supplementary Figure 1a, it is observed that the quintuple mutant, with disruptions in five genes (*dmsr-4*, *dmsr-5*, *dmsr-6*, *dmsr-7*, and *dmsr-8*), exhibited defects in the O₂-regulated locomotor behavior. However, the authors did not provide an explanation for their focus on the three mutations (*dmsr-4*, *dmsr-7*, and *dmsr-8*) or whether *dmsr-5* and *dmsr-6* are also involved in the hypoxia-evoked locomotory response. Clarification regarding these aspects is necessary to enhance the understanding of the study findings.

Our apology for inadvertently omitting some data in our initial submission, which led to this confusion. We have now included the data showing that *dmsr-5; 7; 8* triple, *dmsr-6; 7; 8* triple, *dmsr-4, 5; 6; 7* quadruple, and *dmsr-4, 5; 6; 8* quadruple mutants had largely normal responses to acute hypoxia (Supplementary Fig. 1e–1h). These data excluded the potential involvement of DMSR-5 and DMSR-6 in hypoxia evoked locomotory responses. These findings, together with the other data in Supplementary Fig. 1a–1d, provided the rationale for our continued investigation of DMSR-4, DMSR-7, and DMSR-8 (Lines 171–173).

3. In figure 5c, whether there are significant differences between the *flp-19;flp-20* mutant and the *frpr-9;flp-19;flp-20* mutant, as well as between the *frpr-19* mutant and the *frpr-9;flp-19;flp-20* mutant?

As suggested by our reviewer, we re-analyzed the data presented in Fig. 5c. There is no significant difference between *flp-19; flp-20* double and *frpr-9; flp-19; flp-20* triple in chemotaxis to 2,3-pentanedione (Fig 5c) (Lines 308–309). In addition, the difference between *frpr-9* and *flp-19* is not significant (Fig 5c), and the defect of *frpr-9; flp-19* double mutants was not significantly different from that of *flp-19* or *frpr-9* single mutants. These observations are consistent with our hypothesis that FRPR-9 forms a physiologically relevant pair with FLP-19 (Lines 305–308)

FLP-20 appears to act in parallel to FRPR-9/FLP-19 signaling. The *flp-19; flp-20* double mutants had a significantly decreased chemotaxis when compared to either *flp-19* or *flp-20* single mutants (Fig. 5c; Supplementary Fig. 4c). Consistently, the chemotaxis of *frpr-9; flp-19; flp-20* triple mutants was significantly lower than that of *frpr-9; flp-19* double mutants or *flp-19* single mutants (Fig. 5c). Therefore, *frpr-9/flp-19* mutants exhibited a synergistic effect when combined with *flp-20* mutants (Fig. 5c; Supplementary Fig. 4c). However, the difference between *frpr-9* single and *frpr-9; flp-19; flp-20* triple mutants did not reach significance even though the response of *frpr-9* strain was higher than that of the triple mutants. This is likely caused by the large variation of the chemotaxis data in *frpr-9* mutants. Nonetheless, we are confident that FRPR-9/FLP-19 likely act together, but in parallel to FLP-20, to regulate AWC-mediated chemotaxis. We hope that our reviewer's concern has been addressed in the new text (Lines 305–319).

4. In Figure 5g and h, it is intriguing to observe that the expression of *flp-19* alone was able to rescue the defect in 2,3-pentanedione chemotaxis observed in *flp-19; flp-20* double mutants. The authors should thoroughly discuss the potential mechanisms underlying this phenomenon.

We apologize for the confusion caused by the insufficient text in our initial submission. Please also see our responses to point 3. The defect of *flp-20* single mutants is subtle and is not significantly different from that of the wild type (Fig 5c; Supplementary Fig. 4c). FLP-20 does not seem to be required for 2,3-pentanedione chemotaxis in the presence of FRPR-9/FLP-19 signaling. We performed additional assays during the revision and confirmed that *flp-20* single mutants had normal response in chemotaxis to the diluted 2,3-pentanedione. We did not include this data since we already have two figure panels showing this (Fig 5c; Supplementary Fig. 4c). Disrupting FRPR-9/FLP-19 signaling reduced but did not eliminate the ability of animals to direct chemotaxis to 2,3-pentanedione. FLP-20 appears to be responsible for a fraction of the residual chemotaxis in animals deficient in FRPR-9/FLP-19 signaling. This observation resembles the attractive response to gustatory cues, which is primarily mediated by ASE neurons. The ADF, ASG and ASI sensory neurons are not required for gustatory responses when ASE neurons are present, while contributing to the residual responses to gustatory cues when ASE neurons are ablated (Bargmann and Horvitz, 1991). Because of the minor role of FLP-20 in chemotaxis, the restoration of FRPR-9/FLP-19 signaling in *flp-19; flp-20* double mutants is expected to fully rescue the chemotaxis defects observed in the double mutant animals. We clarified this in the revised manuscript in Lines 305–319.

5. In Figure 6g, the authors demonstrated that overexpressing *srx-1* and *srx-3* in the AWCOFF neuron successfully rescued the reduced chemotaxis observed in *srx-2* mutants, although SRX-2 appeared to play a more prominent role. Interestingly, the authors also noted that the expression levels of *srx-1* and *srx-3* were significantly lower in AWCOFF compared to *srx-2*. This raises the question of whether the higher expression level of *srx-2* determines its major role. The authors should discuss the potential relevant functions of the three proteins in more detail. Additionally, it is important to address why the AWCOFF neuron requires three genes encoding 2,3-pentanedione receptors and the significance of this observation.

We thank our reviewer for the comments 5 and 6. We jointly address these two comments because of their connections.

In our initial submission, we did not examine if ectopic expression of *srx-1* or *srx-3* in AWB neurons is sufficient to generate aversive responses to the diluted 2,3-pentanedione. We now show that animals with the mis-expression of *srx-1* or *srx-3* in AWB neurons are repelled by the diluted 2,3-pentanedione (Supplementary Fig 6f), supporting that SRX-1 and SRX-3 are putative receptors for 2,3-pentanedione. Interestingly, the aversive response of these animals is not as pronounced as in animals with ectopic expression of *srx-2* in AWB neurons (Supplementary Fig 6f). This observation implies that SRX-2 may have a higher affinity to 2,3-pentanedione or have a higher efficiency in initiating the relevant responses. This was confirmed in the single worm chemotaxis assay, showing that mis-expression of *srx-1* in AWB neurons was less effective in triggering an aversive response compared to *srx-2* (Supplementary Fig 6g). The co-expression of *srx-1* and *srx-2* in AWB neurons results in an aversive response that closely resembles the response seen in animals with ectopic *srx-2* expression in AWB neurons (Supplementary Fig 6g). This further demonstrated that SRX-2 is likely the primary receptor for low concentrations of 2,3-pentanedione, and dominates the behavioral response to this dilution of 2,3-pentanedione (Lines 373–387).

Because three receptors exhibit different levels of expression, odorant affinity or efficiency in eliciting behavioral responses, we postulate that a different receptor might have a predominant role in a specific context, dependent on the odor concentrations present in the surroundings, which is similar to the involvement of multiple neurons in response to different

concentrations of NaCl (Bargmann and Horvitz, 1991). The presence of three receptors with different express levels and different affinities might increase the efficiency and fidelity for *C. elegans* in response to varying concentrations of 2,3-pentanedione. It might also enhance the adaptability of wild *C. elegans* in a dynamic environment, and allow animals to maintain their ability to locate food or avoid dangers in the face of changes in their surroundings (Lines 373–387).

6. Whether ectopic expression in AWB together with *srx-1* and *srx-2* can increase the aversive responses to 2,3-pentanedione?

See our response to the point 5 and the description in Lines 373–387 of the main text.

Minor:

1. In figure 2c, the authors stated that by examining the responses of each single mutant, we found that mutating *flp-1* yielded a phenotype similar to that of the quadruple mutants. It is better to include the data of other single mutants (*flp-14*, *flp-23*, and *flp-25*) in the supplementary figures.

We have included the acute hypoxia responses of each single mutant (*flp-14*, *flp-23*, or *flp-25*) in Supplementary Fig. 1k, 1l, and 1m.

2. In Supplementary Figure 2a and b, the different colors represent distinct categories of genes. However, in Supplementary Figure 2c and d, the same set of colors represents different levels of significance. This can potentially lead to confusion.

We thank our reviewer for this. Different colors are now used in Supplementary Fig. 2a, b and in Supplementary Fig. 2c, d.

3. In Figure 4e, it would be beneficial to include information about the specific chemosensory neurons that are involved in detecting individual chemicals. This additional information will provide a more comprehensive understanding of the neuronal basis for the observed responses.

As suggested by our reviewer, we have included the relevant information in Fig. 4e.

4. In line 318, “AWC mediated” should be “AWC-mediated”.

We have fixed this.

References:

- Bargmann, C.I., and Horvitz, H.R. (1991). Chemosensory neurons with overlapping functions direct chemotaxis to multiple chemicals in *C. elegans*. *Neuron* 7, 729-742.
- Chalasani, S.H., Kato, S., Albrecht, D.R., Nakagawa, T., Abbott, L.F., and Bargmann, C.I. (2010). Neuropeptide feedback modifies odor-evoked dynamics in *Caenorhabditis elegans* olfactory neurons. *Nat Neurosci* 13, 615-621.

Choi, W., Ryu, S.E., Cheon, Y., Park, Y.J., Kim, S., Kim, E., Koo, J., Choi, H., Moon, C., and Kim, K. (2022). A single chemosensory GPCR is required for a concentration-dependent behavioral switching in *C. elegans*. *Current Biology* 32, 398-+.

Meisel, J.D., Panda, O., Mahanti, P., Schroeder, F.C., and Kim, D.H. (2014). Chemosensation of bacterial secondary metabolites modulates neuroendocrine signaling and behavior of *C. elegans*. *Cell* 159, 267-280.

REVIEWERS' COMMENTS

Reviewer #1 (Remarks to the Author):

The authors have effectively revised their manuscript to address quite a long list of specific criticisms from the three reviewers.

My principle criticism was that, in the initial version of the manuscript, the abstract and some aspects of the Introduction were just not very clearly written. The authors have completely revised the abstract and parts of the Introduction. I find the revised Introduction very good. The revised Abstract is much improved, and although some mildly awkward wording remains, it is acceptable in the revised form.

The authors responded to 16 additional specific criticisms I had of the original manuscript. I have gone through each of these carefully and am satisfied with each response.

I also went through, albeit less carefully, the authors' responses to the specific criticisms of the other two reviewers. These responses also appear to be satisfactory from my point of view.

Reviewer #2 (Remarks to the Author):

My concerns have been fully addressed, and I think that the revised manuscript fits well to Nature Communications.

Reviewer #3 (Remarks to the Author):

The authors have satisfactorily addressed all of my concerns, and I have no further requests.